# Progressing Towards the Sustainable Development of Cream Formulations

**DOI:** 10.3390/pharmaceutics12070647

**Published:** 2020-07-09

**Authors:** Ana Simões, Francisco Veiga, Carla Vitorino

**Affiliations:** 1Faculty of Pharmacy, University of Coimbra, 3000-548 Coimbra, Portugal; simoesana88@gmail.com (A.S.); fveiga@ff.uc.pt (F.V.); 2Associated Laboratory for Green Chemistry of the Network of Chemistry and Technology (LAQV/REQUIMTE), Group of Pharmaceutical Technology, Faculty of Pharmacy, University of Coimbra, 3000-548 Coimbra, Portugal; 3Coimbra Chemistry Center, Department of Chemistry, University of Coimbra, 3004-535 Coimbra, Portugal; 4Centre for Neurosciences and Cell Biology (CNC), Faculty of Medicine, University of Coimbra, 3004-504 Coimbra, Portugal

**Keywords:** topical dermatological product, cream formulation, quality by design, Box-Behnken design, microstructure, rheology, performance

## Abstract

This work aims at providing the assumptions to assist the sustainable development of cream formulations. Specifically, it envisions to rationalize and predict the effect of formulation and process variability on a 1% hydrocortisone cream quality profile, interplaying microstructure properties with product performance and stability. This tripartite analysis was supported by a Quality by Design approach, considering a three-factor, three-level Box-Behnken design. Critical material attributes and process parameters were identified from a failure mode, effects, and criticality analysis. The impact of glycerol monostearate amount, isopropyl myristate amount, and homogenization rate on relevant quality attributes was estimated crosswise. The significant variability in product droplet size, viscosity, thixotropic behavior, and viscoelastic properties demonstrated a noteworthy influence on hydrocortisone release profile (112 ± 2–196 ± 7 μg/cm^2^/√h) and permeation behavior (0.16 ± 0.03–0.97 ± 0.08 μg/cm^2^/h), and on the assay, instability index and creaming rate, with values ranging from 81.9 to 120.5%, 0.031 ± 0.012 to 0.28 ± 0.13 and from 0.009 ± 0.000 to 0.38 ± 0.07 μm/s, respectively. The release patterns were not straightforwardly correlated with the permeation behavior. Monitoring the microstructural parameters, through the balanced adjustment of formulation and process variables, is herein highlighted as the key enabler to predict cream performance and stability. Finally, based on quality targets and response constraints, optimal working conditions were successfully attained through the establishment of a design space.

## 1. Introduction

In dermatological therapy, semisolid dosage forms, including cream formulations, remain the gold-standard vehicles for topical drug delivery [1]. Topical therapeutic efficacy is highly dependent on skin conditions, physicochemical properties of the active substance and vehicle/formulation characteristics because of their significant impact on drug release and permeation. Besides stratum corneum (SC) barrier function, structural changes of diseased skin, active substance solubility, lipophilicity, molecular weight, concentration and physical state (solubilized or dispersed), the understanding and selection of a suitable vehicle microstructure is of crucial importance, since it plays a fundamental role on skin application/sensory properties, formulation appearance, product performance, physical stability and patient compliance.

Cream formulations are described as multiple-phase systems, particularly susceptible to instability phenomena. Indeed, mixing and interactions among multifunctional excipients (e.g., emulsifying agents, thickeners, long-chain fatty acids or alcohols and preservatives) and active substance, along with manufacturing process parameters produce important modifications on relevant vehicle microstructure (droplet size, rheological properties, homogeneity, pH and polymorphism), conferring different cream physicochemical properties [2,3,4,5]. Only an integrated approach will enable to design cream formulations in a more sustainable manner. But, what is the interdependency among variables and how can we systematically measure and control it?

First, the effect of formulation- and process-related variables on vehicle microstructure feature must be thoroughly inspected [6].

Second, it is demanded to understand the extension of these effects on product performance. To this end, in vitro release (IVRT) and permeation testing (IVPT) is mandatory. IVRT yields the drug release rate and kinetics, which is a result of the diffusion mechanism governed by vehicle-drug interactions, while IVPT renders the ability and extension of drug penetration throughout skin, which rely on drug properties, and a joint contribution of drug-skin and vehicle-skin interactions. According to the dosage form, formulation and process parameters, differences on IVRT and IVPT responses, herein considered critical quality attributes (CQAs) are expected, which are intrinsically linked to discriminatory power of the methodologies. [6,7,8,9,10,11,12,13,14].

Third, monitoring instability mechanisms deeming from chemical (pH), physical (drug and vehicle non-homogeneity) and microbiological changes that may occur during the manufacturing and shelf-life conditions, and eventually destroying formulation microstructure, is another major concern [15]. For that, specific and stability-indicating tests should likewise be established.

When envisioning an accurate and robust product, with maximized performance, the optimized formulation and manufacturing conditions must be established through groundbreaking methodologies. Accordingly, regulatory authorities are encouraging pharmaceutical industry to implement a more systematic and scientific-based approach in the early stages of pharmaceuticals design and development. In this context, the application of Quality by Design (QbD) is introduced as an opportunity to improve product and manufacturing robustness, efficiency and productivity, with substantial reduction in time and cost production, product variability and batch rejection. Such concept fuels an in-depth understanding about the impact of variability sources on product quality attributes. More acquired information will provide more control, also supporting regulatory flexibility [16,17,18,19].

The implementation of QbD concepts begins with the quality target product profile (QTPP) determination, and the critical quality attributes (CQAs) identification. A risk assessment is carried out in order to identify and prioritize the critical material attributes (CMAs) and critical process parameters (CPPs) that potentially affect product CQAs. Subsequently, a design of experiment (DoE) is performed to determine the functional relationship among CMAs and CPPs, and the product CQAs. Finally, optimal operating ranges to CMAs and CPPs are established within a design space (DS) [17,20].

The main goal of this study is to develop an optimal cream formulation, and simultaneously providing a useful guideline for topical pharmaceutical manufacturers. For that purpose, a tripartite analysis, including microstructure, IVRT-IVPT, and stability evaluation, was set-forth through the implementation of QbD methodology. Measuring the extent of CMAs and CPPs impact on cream CQAs will ensure to yield a consistent quality product which met QTPP specifications [21,22,23].

For such a purpose, a commercially available 1% hydrocortisone (HC) cream formulation was used as reference. HC cream QTPP and CQAs were initially identified. Furthermore, foregoing screening outcomes and the current risk analysis, a Box-Behnken design was performed to scrutinize how CMAs and CPP impact cream quality attributes, exploring the interplay of microstructure CQAs and product performance metrics. Therefore, a detailed microstructure characterization, encompassing droplet size and rheological profile, as well as in vitro release and permeation behavior was carried out. Formulation stability was also assessed. Finally, considering statistical DoE data analysis and quality requirements, the best working conditions were identified through the establishment of a DS.

## 2. Materials

Micronized hydrocortisone was kindly provided by Laboratórios Basi - Indústria Farmacêutica S.A. (Mortágua, Portugal). Methyl parahydroxybenzoate and propyl parahydroxybenzoate were purchased from Alfa Aesar (Kandel, Germany). Kolliwax^®^GMS II (glycerol monostearate), Kolliwax^®^CA (cetyl alcohol), Kollicream^®^IPM (isopropyl myristate) and Dexpanthenol Ph. Eur. were kindly provided by BASF SE (Ludwigshafen, Germany). Stearic acid was provided by Acorfarma Distribuición S.A. (Madrid, Spain). Triethanolamine was purchased from Panreac AppliChem (Darmstadt, Germany). Liquid paraffin was provided by LabChem Inc (Zelienople, Pennsylvania). Glycerol was purchased from VWR Chemicals (Leuven, Belgium). Water was purified (Millipore^®^) and filtered through a 0.22 μm nylon filter before use. All other solvents were analytical or high-performance liquid chromatography (HPLC) grade.

## 3. Methods

### 3.1. Quality by Design Approach

#### 3.1.1. Definition of QTPP

The QTPP was established, prospectively comprising certain cream quality features that ideally should be reached, taking into account drug product efficacy and safety.

#### 3.1.2. Identification of CQAs

Potential CQAs were identified as a set of QTPP that should be within an appropriate limit to ensure cream quality achievement.

#### 3.1.3. Initial Risk Assessment

To identify CMAs and CPPs, a Failure Mode, Effects and Criticality Analysis (FMECA) was constructed to quantify the risk or failure mode(s) associated with each formulation and/or process parameter and to assess their impact on cream CQAs.

Risk quantification was performed considering the severity (S), probability of occurrence (P) and detectability (D) of each parameter using a numerical scale from 1 to 5, with 1 being the lowest severity, probability and undetectability and 5 the highest. For each factor, the rank and prioritization of the risk was conducted according to the risk priority number (RPN) given by RPN = S × P × D. The factors presenting higher RPN values were subjected to a further optimization process.

#### 3.1.4. DoE

To statistically optimize the HC cream formulation and manufacturing process, a three-factor, three-level Box-Behnken design was performed, using JMP 14.0 Software (Cary, USA). Such a design is a suitable DoE for exploring quadratic response surfaces and constructing second-order polynomial models. According to preliminary studies and risk assessment analyses, glycerol monostearate amount (x_1_), isopropyl myristate amount (x_2_) and homogenization rate (x_3_) were recognized as the most significant factors affecting cream CQAs and were varied at high (+1), medium (0) and low (−1) levels. The selection of range factors was determined based on previous experimental results [1]. A total of fifteen runs, with three center points, were generated. Different coded level combinations are described in Table 1, while DoE runs are presented in Table 2. Experiments were randomly carried out.

The effects of independent variables on different responses/CQAs were investigated using the following non-linear quadratic model (1):*Y_n_* = *β*_0_ + *β*_1_*x*_1_ + *β*_2_*x*_2_ + *β*_3_*x*_3_ + *β*_12_*x*_1_*x*_2_ + *β*_13_x_1_*x*_3_ + *β*_23_*x*_2_*x*_3_ + *β*_11_*x*_1_^2^ + *β*_22_*x*_2_^2^ + *β*_33_*x*_3_^2^(1)
where *Y* denotes the response associated with each factor level combination; *β*_0_ depicts the arithmetic average; *β*_1_, *β*_2_ and *β*_3_ represent the first order coefficients of the respective independent variables; *β*_12_, *β*_23_ and *β*_13_ typify the interaction coefficients; *β*_11_, *β*_22_ and *β*_33_ betokens the quadric coefficients. The positive and negative signs of the coefficient values indicate a synergetic or antagonistic effect of each term, respectively, while the magnitude represents the impact extent.

#### 3.1.5. Optimization Process

Taking into account the fitted model information, along with microstructure and performance requirements (match target), a DS was graphically defined to establish the optimal working conditions of the most important variables.

### 3.2. Preparation of Hydrocortisone Cream Formulations

HC cream formulations were prepared following a conventional manufacturing method, resorting to a high energy emulsification technique, as previously described [1]. Briefly, excipients from the dispersed and the continuous phases were separately dissolved while heating at 70 °C. The temperature of each unit operation was based on raw material melting point, ensuring that all ingredients were in the molten state. When carefully weighed, the micronized HC was solubilized into the oily phase. Depending on the formulation and process design, different amounts of glycerol monostearate (x_1_), and isopropyl myristate (x_2_) were dissolved in the dispersed phase. In a constant volume mixing vessel, an oil-in-water (o/w) emulsion was obtained by adding the dispersed phase dropwise into the continuous phase. The mixture was subsequently homogenized using a high shear rotor-stator mixer (Ultra-Turrax X10/25, Ystral GmbH, Dottingen, Germany), for a total of 15 min at 70 °C. According to factorial design, distinct homogenization rates (x_3_) were also applied. The Ultra-Turrax tip was kept at a constant height. Cream formulations were cooled down at room temperature. Batches of 500 g were produced.

### 3.3. Drug Content

Considering emulsion-based products, separation phenomena may occur during the manufacturing process and shelf life. Therefore, to ensure formulation homogeneity and physical stability, the drug content of the final product was determined. An appropriate amount of accurately weighed cream was removed from the top, middle, and bottom of the container, and transferred to a flask. HC content was extracted and analyzed through reversed-phase high-performance liquid chromatography (RP-HPLC). A Shimadzu LC-2040C 3D apparatus equipped with a quaternary pump, an autosampler unit, and a D2 Lamp UV-visible photodiode array detector was employed. A LiChrospher100 RP-18, 5 μm (4.6 mm × 125 mm) column (Merck KGaA, Germany), with a LiChrospher100 RP-18, 5 μm (4 mm × 4 mm) pre-column (Merck KGaA, Germany), was used for the analysis. The mobile phase consisted of a mixture of acetonitrile-water (75:25, V/V) pumped at a constant flow rate of 0.8 mL/min for 25 min at 30 °C. An injection volume of 10 μL was considered for all standards and samples. The detection was performed at 242 nm [2].

### 3.4. pH

Topical products should present an appropriate pH range, since this may influence drug solubility, stability and potentiate skin irritation. The HC cream’s pH was determined at 25 °C, using a digital pH C3010 Multiparameter Analyzer (Consort bvba, Turnhout, Belgium). The pH meter was calibrated using standard buffer solutions (4.00, 7.00, 10.00). About 1.0 g of each formulation was weighed and dispersed in 10 mL of distilled water, and the respective pH was measured. The determination was performed in triplicate, 24 h after batch manufacturing.

### 3.5. Droplet Size

Emulsions are colloidal dispersions, wherein droplet size is one of the main factors that affect their optical appearance, rheology and physical stability, and consequently their quality profile. A droplet formulation size analysis was carried out using an Eclipse 50i optical microscope (Nikon Instruments Europe BV, Amsterdam, Netherlands), 4 days after batch manufacturing. A minimal amount of each formulation was dispersed on a slide and the cover slip was softly placed to avoid breaking the system structure. Three microscopy images were acquired for each sample, and droplet length was measured (*n* = 30 per image) using imaging software (NIS Elements version 3.10).

### 3.6. Rheological Aspects

Viscosimetric measurements provide noteworthy information regarding formulation, application/sensorial properties and structural stability during shelf life. The rheological behavior of the creams was analyzed using a Haake^TM^ MARS^TM^ 60 Rheometer (ThermoFisher Scientific, Germany) with a controlled temperature maintained by a thermostatic circulator and a Peltier temperature module (TM-PE-P) for cones and plates. Data were analyzed with Haake Rheowin^®^ Data Manager v.4.82.0002 software (ThermoFisher Scientific, Germany). Throughout the experimental analysis, temperature was maintained at 32 °C. For each test, approximately 1.0 g of each formulation was placed on the lower plate before slowly lowering the upper geometry to the predetermined trimming gap of 1.1 mm. After trimming the excess material, the geometry gap was set at 1 mm. Rotational and oscillatory measurements were performed sequentially on each sample for a thorough rheological characterization [3]. Rotational tests enable us to evaluate small periodic deformations that determine breakdown or structural rearrangement and hysteresis, while oscillatory tests allow us to analyze material viscoelastic properties when they are exposed to small-amplitude deformation forces. All rheological studies were performed in triplicate.

#### 3.6.1. Rotational Measurements

Rotational measurements were addressed using cone (P35 2°/Ti; 35 mm diameter, 2° angle) and plate (TMP 35) geometry configuration. Viscosity curves [η = f($\dot{ɣ}$)] were recorded by a control rate flow step from 0.1 to 50 s^−1^ for 10 min. Significant differences among viscosity curves were observed at high shear rate values, which is clearly distinguish in a logarithmic-linear scale representation. Apparent viscosity (*η*_10_, Pa.s) was obtained at a shear rate of 10 s^−1^. Different mathematical models were fitted to the acquired flow curves when searching for the best descriptive model: Ostwald de Waele, Herschel-Bulkley, Bingham, Casson and Cross [4,5,6]. The best fitting was selected, considering the regression coefficient values (R^2^).

Additional flow curves were generated by ramping the shear rate from 0.01 to 300s^−1^ over 3 min (ascendant curve) and then from 300 to 0.01 s^−1^ during 3 min (descendent curve). The thixotropic behavior was estimated by considering hysteresis loop areas (S_R_, Pa/s).

#### 3.6.2. Oscillatory Measurements

Oscillatory measurements were carried out using plate (P20/Ti, 20 mm diameter) and plate (TMP 20) geometry configuration. First, the linear viscoelastic region plateau (LVR, Pa), yield stress (*τ*_0_, Pa) and flow point (*τ_f_*, Pa) were estimated from the amplitude sweep tests, conducted in a shear stress ranging from 1 to 600 Pa, at a constant frequency of 1 Hz. Afterward, the storage modulus (*G*’, Pa), loss modulus (*G*″, Pa) and loss tangent (*tan δ*) were determined from the frequency sweep tests, performed over a frequency range from 100 to 0.1 Hz, at a constant shear stress of 1 Pa [7].

### 3.7. In Vitro Release Studies

IVRT was conducted using static vertical Franz diffusion cells (PermeGear, Inc., Pennsylvania, USA) with a diffusion area of 0.636 cm^2^ and a receptor compartment of 5 mL. A dialysis cellulose membrane (molecular weight cut-off 14,000, avg. flat width 33 mm, D9652-100FT, Sigma-Aldrich), previously soaked overnight in distilled water, was placed between donor and receptor compartments. The receptor medium, a mixture of ethanol-water (30:70), was appropriately screened based on HC solubility studies to ensure the sink conditions during the experiment. The release media was continuously stirred at 600 rpm and maintained at 37 °C by a thermostatic water pump, assuring a temperature of 32 °C at the membrane surface (to mimic skin conditions). All tests were conducted for 24 h. In total, 300 mg of each formulation was evenly spread over the membrane surface. The donor compartment and the receptor sampling arm were carefully covered with Parafilm^®^ to avoid unnecessary evaporation and to achieve occlusive conditions. Samples of the receptor phase (300 μL) were withdrawn at 0.25, 0.5, 1, 2, 3, 4, 6, 8, 10 and 24 h, and analyzed by RP-HPLC. The same volume of medium was replaced with fresh receptor solution [8,9]. The percentage of HC released into the medium was calculated using the following Equation (2):
(2)$$\mathrm{Cumulative}\;\mathrm{release}\;\mathrm{percentage}=\overset{t}{\underset{t=0}{\sum }}\frac{M_{t}}{M_{0}}\times 100$$
where *M_t_* is the cumulative amount of HC released at each sampling time point, *t* is time and *M*_0_ is the initial weight of the HC in the formulations. The cumulative % of HC released after 6 h (R_6h_) and 24 h (R_24h_) were used for comparison among formulations.

In order to identify the release pattern of HC from the vehicle, release data were fitted into two models: Higuchi and Korsmeyer-Peppas [10].

### 3.8. In Vitro Permeation Studies

IVPT was performed in static vertical Franz diffusion cells, in the same conditions of the IVRT, but using newborn pig skin, clamped between the donor and receptor compartments, with the SC side facing up. Permeation tests were conducted for 48 h. A PBS-ethanol (70:30) solution was considered as receptor medium. Formulations were tested under finite dose conditions. Samples of the receptor phase were withdrawn at 0.5, 1, 3, 6, 10, 24, 30, 36 and 48 h, and analyzed through RP-HPLC.

The full-thickness pig skin was treated with a manual dermatome (BA706R, AESCULAP, Tuttlingen, Germany) which cut a surface-parallel skin layer with a specified thickness. Dermatomed skin or split skin comprises the epidermis, including SC, and portions of the dermis. The exact thicknesses of the split pig skin samples were 0.80 ± 0.16 mm [11]. The split skin sample sheets were cut, wrapped with aluminum foil and stored at −20 °C until used. The storage time for the skin samples was less than 3 months. Prior to the experiments, the frozen skin pieces were thawed, and hydrated by placing in distilled water overnight in a refrigerator (at about 4 °C). Skin integrity was monitored by measuring the transepidermal water loss (TEWL). TEWL values higher than 12 g/m^2^.h were ruled out from the experiment [24].

The cumulative amount of HC diffused per unit area of the excised skin (*Q_n_*) was calculated as a function of time (t, h) according to the following expression (3):
(3)$$Q_{n}=\left(C_{n}\times \;V_{0\;}+\overset{n-1}{\underset{i=1}{\sum }}C_{i}\times V_{i}\right)/A$$
where *C_n_* corresponds to the drug concentration of the receptor medium at each sampling time, *C_i_*, to the drug concentration of the i^th^ sample, *A*, to the effective diffusion area, and *V*_0_ and *V_i_* to the volumes of the receptor compartment and the collected sample, respectively. The cumulative amount of HC (μg/cm^2^) permeated after 6 h (Q_6h_), 24h (Q_24h_) and 48h (Q_48h_) were used for comparison among formulations.

According to Fick’s first law of diffusion, the steady-state flux (J_ss_, μg/cm^2^/h) can be expressed by (4):J_ss_ = DC_0_P/h = C_0_Kp(4)
where D is the diffusion coefficient of the drug in the SC, C_0_ represents the drug concentration in the donor compartment, P is the partition coefficient between the vehicle and the skin, h is the diffusional path length, and K_p_ stands for the permeability coefficient. The flux and K_p_ of the yielded formulations were measured and compared accordingly. The enhancement ratio (ER) for flux was calculated as the ratio between the flux of different formulations and the target flux value. The J_ss_ and K_p_ of the yielded formulations were calculated and compared accordingly. Permeation lag time (t_lag_), a parameter related with the required time to achieve the steady-state flux of a drug through the skin, was also considered for analysis.

The study was conducted in accordance with the Declaration of Helsinki, and the protocol was approved by the Local Ethics Committee.

### 3.9. Stability Analysis

A predictive assessment of the formulation’s physical stability was carried out after 4 days of manufacture using the LUMiSizer equipment (LUM GmbH, Berlin, Germany). This analytical photocentrifugation system measures the transmitted intensity of near-infrared (NIR) light as a function of time and position along the entire sample length. The data are displayed as a function of the radial position, as the distance from the center of rotation (transmission profiles). The shape and progression of the transmission profiles provide the determination of the sedimentation and/or creaming rates, important parameters to assess sample separation phenomena [12,13]. Formulation stability was also quantitatively described through the instability index parameter. This is a dimensionless number and ranges from 0 (more stable) to 1 (more instable). This means that, for the same total clarification, samples with lower clarification rates trend to present more long-term stability. All samples were analyzed in duplicate after 84h of centrifugation conducted at an acceleration of 4000 rpm and 40 °C. Stability parameters were determined using the SEPView^®^ software.

### 3.10. Statistical Analysis

An analysis of variance (ANOVA) was performed using JMP v.14 Software (Cary, IL, USA) to statistically analyze the fitted models. In order to test whether the terms were statically significant in the regression model, Student’s *t*-tests were performed. Statistical analysis is considered significant if the regression Prob > F and *t*-test Prob > |t| are less than 0.05. However, a significant model does not mean a correct explanation of the results variation. The maximum squared regression coefficient (R^2^) indicated how well the model fit the experimental data, and the closer the value is to 1, the better the fit.

Two Fisher tests were also used to assess the adequacy of the model fitting. A regression *F* Ratio (*F*_1_) much larger than 1 suggests a good correlation among the experimental and predicted responses and, therefore, that the regression model is adequate to describe the response variations. In turn, a lack of fit *F* Ratio (*F*_2_) close to 1 indicates the excellent reproducibility of the purchased data (model’s validity). Pure errors, irrespective of the model (e.g., experimental errors), are minimal when a non-significant lack of fit is verified. Thus, a model will be satisfactory when the regression is significant and a non-significant lack of fit is obtained for the selected confidence level [12,14,15].

## 4. Results and Discussion

### 4.1. Definition of QTPP and CQAs Identification

In a QbD-based development approach, pharmaceutical products should be designed according to stakeholders’ requirements (patient expectations, industrial and regulatory aspects) [16]. Taking into account such considerations and preliminary studies, the QTPP profile was predefined for a HC cream formulation (Table 3) [17,18].

Thereafter, based on QTPP, CQAs presenting the highest probability to generate a product failure were properly identified and justified in Table 3. Such a list also comprises individual specifications and a rationale for the selection. For this purpose, droplet size, apparent viscosity (*η*_10_), hysteresis loop area (S_R_), linear viscoelastic region (LVR) plateau, yield stress (*τ*_0_), flow point (*τ_f_*), loss modulus (*G’*), storage modulus (*G″*), loss tangent (*tan δ*), release rate constant of Higuchi model (c_1_), diffusion release exponent of Korsmeyer-Peppas model (c_2_), cumulative % of HC released after 6 h and 24 h (R_6h_ and R_24h_), flux at steady state (J_ss_), permeability coefficient (K_p_), cumulative amount of HC permeated after 6 h, 24 h and 48 h (Q_6h_, Q_24h_ and Q_48h_), pH, assay, instability index, sedimentation rate and creaming rate were acknowledged as the quality attributes most threatened by formulation and process variability and, for that reason, were further investigated.

### 4.2. Initial Risk Assessment

Scientific understanding of how formulation and/or process parameters influence product CQAs is extremely important for risk mitigation. Through a risk analysis, critical sources of product variability should be identified and analyzed in the early stages of product development, and repeated as more knowledge is generated [25]. Therefore, based on acquired data and knowledge, the impact severity of each failure along with the probability of occurrence and detectability was evaluated, and critical parameters were identified [26]. Represented in Table 4, a FMECA was constructed to estimate the risk associated with each formulation- and process-related factor variation. In such a representation, failure modes, causes and effects were also summarized. A cut-off value of RPN above 40 was established for discriminating the important factors (high risk) from nonimportant ones (low risk). With RPN values of 48, 45 and 40, glycerol monostearate amount (x_1_), isopropyl myristate amount (x_2_) and homogenization rate (x_3_) were considered the higher risk factors, while other ones were distinguished as moderate or low risk levels. Such results are in agreement with previous screening outcomes [1].

### 4.3. Scrutinizing DoE

The main challenge in topical corticosteroid therapy is to enhance product efficacy, by increasing the active ingredient’s bioavailability at keratinocytes, fibroblasts and immune cells within the viable epidermis and dermis, without increasing its concentration, and local or systemic side effects [27]. According to the formulation composition and manufacturing process, a semisolid product may present differences in drug content uniformity and physicochemical vehicle properties, with a significant impact on its performance [28,29,30,31].

A great understanding about the effect of formulation and process variability on cream CQAs is thus desirable to establish the best experimental conditions for the optimal product performance. To this end, a multivariate optimization strategy was herein used, supported by a two-step experimental set-up, comprising (i) screening (full factorial, fractional-factorial, Plackett-Burman) and (ii) optimization (central composite, Box-Behnken, Doehlert and D-optimal) designs [32].

In our previous experiments, a two-level Plackett-Burman design was already performed to screen the most influential factors [1]. In the current study, optimization through Box-Behnken design is presented. Box-Behnken is a simple model that, with a minimal number of experiments, allows for the estimation of the main effects by fitting a polynomial model of multiple linear regression [33,34]. Specifically, the impact of x_1_, x_2_ and x_3_, and their interactions on predefined CQAs were simultaneously studied. At different factor level combinations, a total of fifteen formulations were produced. To evaluate the effect of those variables on HC cream CQAs, DoE formulations were characterized for the main quality attributes (Table 5, Table 6, Table 7, Table 8 and Table 9). The collected experimental data were analyzed, and a second-order polynomial model was fitted. The adequacy and significance of each model are summarized in Appendix A. The coefficient values and corresponding significance levels were also included (Appendix A). For better visualization, the main interaction effects were represented through 3D response surface plots.

An overview of the fitted models indicates that glycerol monostearate amount (x_1_), homogenization rate (x_3_) and isopropyl myristate amount (x_2_) demonstrate a decreasing influence on formulation microstructure, performance and stability. The combinatorial analysis pointed out F_8_ as the optimal formulation, since that formulation met specifications pre-established in the QTPP profile.

Droplet size, rheological profile, IVRT and IVPT results, assay and creaming rate were the major impacted CQAs, since significant variations were observed at different experimental conditions.

pH, instability index and sedimentation rate were considered the minor impacted CQAs, once they do not present significant variations at different factor level combinations.

In general, it is possible to infer that formulation parameters impose higher variability than process parameters.

A tripartite analysis will be conducted following the assumptions described in the “Draft guideline on quality and equivalence of topical products” [23].

#### 4.3.1. Statistical Analysis

As shown in Appendix A, the regression data demonstrates that the fitted models for droplet size, *η*_10_, S_R_, LVR plateau, *τ*_0_, *G’, G″*, pH, instability index and creaming rate responses present statistical significance (Prob > F < 0.05), highlighting the importance of the terms on the considered CQAs. In turn, the fitted models for *τ_f_*, *tan δ*, c_1_, c_2_, R_6h_, R_24h_, J_ss_, K_p_, Q_6h_, Q_24h_, Q_48h_, assay and sedimentation rate responses were not statistically significant (Prob > F > 0.05), indicating the nonimportance of the terms or an inadequate fit.

The regression coefficients (R^2^ > 0.8) also demonstrated that the quadratic model is an adequate fit to represent droplet size, *η*_10_, S_R_, LVR plateau, *τ*_0_, *τ_f_*, *G’*, *G″*, *tan δ*, c_2_, R_24h_, assay, pH, and instability index and creaming rate responses, enabling the good predictive power of the considered factors.

Non-significant lack of fit (Prob > F > 0.05) suggests that the fitted mathematical models demonstrated a great ability in the prediction of the following statically significant responses: droplet size, *η*_10_, S_R_, *τ*_0_, pH, instability index and creaming rate.

Regression and thes lack of fit *F* Ratio (*F*_1_ >> 1 and *F*_2_ close to 1) suggest a good correlation among the experimental and the predicted response. Therefore, the regression models of droplet size, *η*_10_, *τ*_0_, pH and instability index are adequate and valid to describe response deviations.

#### 4.3.2. Microstructure

##### Droplet Size

With respect to formulation and process variability impacts on the cream microstructure, the oil droplet size varied from 1.40 ± 0.28 (F_15_) to 3.2 ± 1.0 μm (F_5_) (Table 5). At different factor level combinations, we found significant differences in droplet size results (*p* < 0.05).

As represented in Appendix A, x_1_, x_3_ and x_2_x_3_ were the most influencing terms (Prob >|t| < 0.05). Coefficient values reveal a synergetic impact of x_1_ term, and an antagonistic effect of x_3_ and x_2_x_3_ terms on the considered CQA.

Droplet size is highly dependent on the dispersed phase volume fraction conferred by the concentration in oily components. Hence, at high levels of x_1_, greater globules are formed, since a larger dispersed phase volume fraction privileges the aggregation process rather than breakage [35]. Higher volume fraction results in an increase in the drag force between the dispersed and the continuous phases, and therefore less turbulence in the vessel, hampering the droplet breakdown [36,37]. Data experiments also support the fact that the droplet sizes increase when a rise in dispersed phase viscosity is verified, since the thickening effect of glycerol monostearate excipient is an obstacle to the breakage process efficiency.

Microscopy analysis also demonstrated a relevant variance in the formulation microstructure, with slightly smaller droplet sizes being achieved at high levels of x_2_ and x_3_ (Figure 1).

Considering the isopropyl myristate, it can be observed that higher x_2_ levels tend to produce smaller droplet sizes, since the reduction in the oil melting temperature along with the dispersed phase viscosity contribute to a more efficient breakage process [38].

Moreover, reduced particle sizes are also achieved when higher homogenization rates (x_3_) are applied, once breaking low viscosity systems requires less mechanical energy than breaking systems with higher viscosity. Such behavior is consistent with previous experiments, wherein an homogenization rate of 22,000 rpm efficiently disrupted the oil globules into smaller sizes [39]. However, a balance between globule size and system viscosity should be established. An increase in the homogenization speed of formulations presenting low viscosity results in larger droplet sizes, since the rise in the droplet collision rate favors the coalescence process. Thereby, we could infer that the homogenization efficiency is closely related to the system viscosity.

In the industrial systems, it is possible to accurately predict the emulsion droplet size for a range of shear rates, mixing geometries, interfacial tensions and viscosities [40]. High-shear mixers are extensively used in intensive energy processes, such as mixing and homogenization. Due to their high rotor speeds, high shear rates, highly localized turbulence dissipation rates, and the narrow spacing between the rotor and the stator, this equipment is able to efficiently disrupt the dispersed phase until small droplets are formed [37].

Resorting to high shear rotor/stator mixers for phase mixing and homogenization procedures, the droplet diameter of liquid-liquid dispersions was optimized in a previous study through a suitable selection of mixer height, time and speed (energy/turbulence dissipation rate) [1]. At a pre-established time, the breaking rate is strongly correlated with the underlying turbulent stress, once significant increments in the mixer speed will cause a considerable elevation in the localized energy dissipation rate, improving the efficiency of the breakage process [41]. Therefore, an increase in the mixer energy dissipation rate produced a decrease in the mean droplet size.

On the other hand, at lower interfacial tension, a better equipment performance and smaller droplets are attained. In liquid-liquid systems, droplet breakage is given by the ratio among the disrupting forces, due to the turbulence, and to the restoring forces, due to interfacial tension. If localized energy dissipation rate forces overwhelm the interfacial tension, droplets break into smaller sizes, whereas, if the interfacial tension is predominant, a resistance to the breakage process will be verified. A lower interfacial tension will reduce the break-up resistance. Hence, a reduced droplet size is also the result of an appropriate emulsifying agent concentration [1,42,43].

#### 4.3.3. Rheological Characterization

##### Rotational Measurements

In terms of formulation flow behavior, apparent viscosity, rheological models and thixotropic behavior were considered.

##### Apparent Viscosity

When inspecting the effect of formulation and process variables on cream viscosity, apparent viscosity at a shear rate of 10 s^−1^ (η_10_) ranged from 0.91 ± 0.11 (F_1_) to 11.1 ± 0.9 Pa.s (F_2_) (Table 6). As displayed in Figure 2, at different factor level combinations, significant differences were observed for the considered CQAs (*p* < 0.05).

As displayed in Appendix A, the glycerol monostearate amount (x_1_) was found to have the most relevant effect (Prob > |t| < 0.05) on the considered CQA. At high levels of x_1_, formulations presented an increment in the apparent viscosity results, showing an expected thickener concentration dependence (Figure 2).

In the current work, the DoE cream formulation was structured by a viscous lamellar gel network phase formed by, apart from stearic acid and cetyl alcohol, an interaction of triethanolamine stearate and different amounts of glycerol monostearate. Triethanolamine stearate is an anionic surfactant contributing to the lamellar phase arrangement with an extensive swelling [1,44,45]. Glycerol monostearate is a fatty amphiphile nonionic ester of glycerol alcohol and stearic acid widely used in pharmaceutical products as thickener, emulsifier, and emollient [46,47]. Due to the similar molecular geometry of triethanolamine stearate and glycerol monostearate, when blended, those molecules are closely packed together contributing to a firm and strength gel network formation [48]. Thereby, at high glycerol monostearate amounts, as the dispersed phase volume fraction increases, an increment of lamellar gel structures in the continuous phase is attained, resulting in more viscous systems.

In o/w cream formulations, emulsifiers are capable of performing a number of functions, either alone or in combination with other formulation excipients. In pharmaceutical emulsions manufacturing, the excess of emulsifier, above the optimal surface coverage, may lead to bridging and, eventually, droplet coalescence, resulting in further growth, which imparts its rheological properties [48]. In addition, the combination of ionic or nonionic emulsifiers and fatty amphiphiles (dispersed phase) may interact with water (continuous phase) to form a swollen lamellar gel network in the continuous phase, where the oily droplet will be entrapped [49,50]. Fatty amphiphiles present a polar head group in their alkyl chains and can pack together into an ordered bilayer structure via hydrogen bonding between the polar head groups, and van der Waal forces of attraction between the nonpolar moieties. At elevated temperatures, a significant amount of water may penetrate and swell into the interlamellar space to form a lamellar liquid crystalline phase (disorder states). Upon cooling, a highly viscous gel network phase (ordered state) is formed with a marked increase in continuous phase viscosity. Thereby, oily droplets will be surrounded by alternating amphiphilic bilayers and interlamellar water layers, in a viscous multilamellar rearrangement [50,51,52].

This combination has an important role in the emulsification process, since it enables the stabilization of the oily droplets during manufacturing by the formation of an interfacial film between the dispersed and the continuous phases. It will confer long term formulation stability, since it prevents against droplet movement and coalescence by structuring the continuous phase. Moreover, lamellar phase component control allows us to manipulate the cream’s rheology. Systems with a similar type and amount of emulsifiers and fatty amphiphiles have similar structures (lamellar gel network) and rheological properties [53].According to gel network theory, the overall cream viscosity relies on emulsifier and fatty amphiphile types and concentrations, and on swelling behavior. Swelling properties are highly dependent on electrostatic repulsive interactions. In lipid membranes, the electrostatic repulsion enables us to incorporate significant amounts of water in the interlamellar space, once hydrogen bonds are promoted near to the hydrophilic groups, and thus phenomenal swelling, inducing changes in rheological properties [51,54]. With the increasing amphiphilic molecule concentration and thus the dispersed phase volume ratio, more gel networks are structured, since more hydrophilic groups are available to bind water molecules. Therefore, a decrease in free water molecules in the continuous phase is verified, causing the system to thicken [53,55,56]. The apparent viscosity results provide predictive information concerning the formulation resistance to structural breakdown, since more structured networks present more resistance to shear-induced deformation [49].

##### Rheological Modeling

As displayed in Figure 2, all formulations exhibited a nonlinear relationship between the shear rate and the shear stress, demonstrating a non-Newtonian, pseudoplastic behavior with decreasing apparent viscosity as the shear rate is increased [57]. This occurs when the oily droplets of an o/w emulsion are deformed into ellipsoidal shapes and start to form layers with the same plane of the shear, offering less resistance to flow. In rotational tests, the flow field will change the molecule orientation to make it parallel to the flow direction, resulting in lower frictional resistance and apparent viscosity. The intermolecular interactions may be diminished due to the microstructural anisotropy as a result of the shear deformation [4,5,58]

For a complete flow behavior characterization of the different DoE formulations, several mathematical models were fitted to the experimental data. Ostwald de Waele (*τ = K.$\overset{˙}{ɣ}$^n^*) is a flow curve model function to describe the shear-thinning and shear-thickening flow behavior of samples without *τ*_0_. Herschel-Bulkley (*τ = τ*_0_
*+ K.$\overset{˙}{ɣ}$^n^*), Bingham (*τ = τ*_0_* + K.$\overset{˙}{ɣ}$*) and Casson (*τ^1/2^ = τ*_0_*^1/2^ + (K.$\overset{˙}{ɣ}$)^1/2^*) functions are, likewise, flow curve models to characterize the shear-thinning and shear-thickening flow behavior of samples presenting *τ*_0_.

The yield point (*τ*_0_) is defined as the minimum shear stress that must be applied to induce material flow. Once exceeded, the formulation will show a structural breakdown. Spreadability is a critical sensory property for topical dosage forms highly dependent on *τ*_0_. Thereby, this parameter is an essential CQA for patient acceptance. For the same formulation, the deviation in *τ*_0_ values among the different rheological models is predictable, since *τ*_0_ determination depends on the rheological method and model function.

Cross function (*η = η_∞_ + [(η_0_ − η_∞_)/(1 + (C.$\overset{˙}{ɣ}$)^m^]*) is a viscosity curve model to characterize samples’ flow behavior based on predictions of the zero shear (*η*_0_) and infinite shear (*η_∞_*) viscosities. This model provides material viscosity details covering three characteristic regions: (i) the first Newtonian plateau range of *η*_0_ value (viscosity function towards an infinitely low shear rate, close to zero); (ii) shear thinning range and (iii) the second Newtonian plateau range of *η_∞_* value (viscosity function towards an infinitely high shear rate).

The significance of each parameter is as follows: *τ* is the shear stress (Pa), *η* expresses the apparent viscosity (Pa.s), $\overset{˙}{ɣ}$ depicts the shear rate (s^−1^) and *K* represents the consistency index (Pa.s^n^); *C* is the Cross time constant (s); *m* is the Cross rate constant.

In case of Ostwald de Waele and Herschel-Bulkley models, *n* refers to the flow behavior index or power-law index (dimensionless). When *n* < 1, the fluid tends towards a shear-thinning/pseudoplastic behavior and the apparent viscosity decreases as the shear rate increases. In turn, if *n* > 1, the fluid tends towards a shear-thickening/dilatant behavior, wherein the apparent viscosity increases with the increase in shear rate, and when *n*=1, the fluid tends towards an ideal viscous flow behavior (Newtonian fluid) [5,6,59,60,61,62,63].

The *K* gives an indication about sample viscosity. It is possible to find a relation between *K* and *n* values. *K* results, for samples with different *n* values, may be properly compared since a modified Ostwald de Waele equation is used [6,60,62,64,65].

Although generally described in the literature, *C* and *m* parameters do not correspond to *K* and *n* terms. In a Cross model, *C* is responsible for the curve shape in the transition from the plateau *η*_0_ to the shear thinning and from the shear thinning to the plateau range of *η_∞_*. The reciprocal, 1/C, is a useful indicator representing the shear rate value required for the shear thinning initiation. *m* (dimensionless) is responsible for the slope of the shear thinning region. In such a model, when 0 < m < 1, the fluid tends towards a shear-thinning/pseudoplastic behavior. In turn, if *m* < 1, the fluid tends towards a shear-thickening/dilatant behavior, and when *m* = 0, the fluid tends towards an ideal viscous flow behavior (Newtonian fluid) [59,61,66].

The regression parameters and terms of each function are represented in Table 7. Considering R^2^ values, the Herschel-Bulkley and Cross models showed an excellent ability for predicting the flow behavior of the different DoE formulations. Herschel-Bulkley model is an extended version of the Ostwald de Waele equation, but comprises a *τ_o_* term and is considered a very useful model to quantitatively describe the shear flow behavior of different materials, including semisolid pharmaceutical products [4]. Accordingly, reaching *τ*_0_, the formulation initiates the flow and, once exceeded, while increasing the shear rate, a decrease in formulation viscosity is observed [67]. In the Herschel-Bulkley model, DoE formulations with higher *η*_10_ tend towards a shear thickening behavior (*n* > 1), while formulations with lower *η*_10_ exhibit shear thinning properties (*n* < 1), suitable for topical administration [68]. A direct relationship between the *τ*_0_ and the *n* index can be also inferred.

In the Cross model, superior viscosity values were denoted when the shear rate is close to zero. In contrast, when the shear rate tends towards infinite values, a decrease in sample viscosity was observed. A direct relationship among formulations *η*_10_, *η*_0_ and *η_∞_* was likewise acknowledged. All DoE formulations exhibit a 0 < *m* < 1 and *η*_0_ > *η_∞_*, mandatory conditions for describing a shear thinning behavior. DoE formulations with lower *η*_10_ tend to present greater *C* values, which result in lower reciprocal results. The lower the system’s viscosity, the lower the shear rate required for shear thinning initiation.

The Bingham and Casson models also provide an amenable description of the formulation flow behavior, which is not seen for the Ostwald de Waele model, since the latter is mainly applied for Newtonian-like systems. A general trend is observed among Herschel-Bulkley, Bingham and Casson models exhibiting superior *τ*_0_ values for the more viscous systems.

##### Thixotropic Behavior

In terms of the effect of formulation and process variables on cream microstructure, the hysteresis loop area (S_R_) varied from 440 ± 80 (F_10_) to 78,470 ± 3401 Pa/s (F_15_) (Table 6). As displayed in Figure 3, at different factor level combinations, significant differences were observed for the investigated CQA (*p* < 0.05).

Thixotropy is a reversible phenomenon exhibited by non-Newtonian materials, characterized by a reduction in the apparent viscosity when the material is subjected to an increased shear force (structure deformation), returning to its original viscosity when shear force decreases (structure reformation) [5]. S_R_ reveals qualitative information toward the breakdown formulation during a deformation and recovery cycle [49,69]

During extrusion from the container, the cream formulation undergoes repeated shear forces. Hence, to prevent structural breakdown, and ensure stability in use, formulation structure recovery must be ensured through a thixotropic behavior. For that reason, this CQA is a good stability indicator [60].

As displayed in Appendix A, glycerol monostearate amount (x_1_) was the most important factor (Prob > |t| < 0.05) for the considered CQA.

It was observed that high levels of x_1_ induced major changes in the DoE formulation microstructure, with flow curves displaying greater S_R_.

The complex rheological behavior of thixotropic materials is understood on the basis of the formulation microstructure, resulting from relatively weak attractive forces among network molecules. During shear rate application, structural deformation is verified (ascendant curve), since intermolecular bonds are weak enough to be broken by the mechanical stresses. When this shear force is diminished, a structural regeneration is observed (descendent curve), with network structure rebuilding [70].

The thickening effect of x_1_ contributed to more viscous systems, with a stronger network and thus a higher resistance of shear-induced deformation. These will require larger shear rates to deform and more time to recover until the initial microstructure is achieved, which could be an undesirable property [62,68,71]. Formulations at medium x_1_ level attained lower S_R_ values, demonstrating their better structure recovery properties.

##### Oscillatory Measurements

Concerning the viscoelastic features, the following responses were considered: linear viscoelastic range (LVR plateau), yield point (*τ*_0_), flow point (*τ_f_*), storage modulus (*G’*), loss modulus (*G″*) and loss tangent (*tan δ*).

The linear viscoelastic range (LVR) is a plateau wherein the storage (*G’*) and loss modulus (*G″*) values are kept constant, meaning that the formulation does not suffer structural breakdown, and represents a material ability in preserving its microstructure when exposed to rising shear stress values. Any deformation will be instantaneously recovered. Thus, the higher the LVR plateau, the higher the formulation microstructure stability [72]. The LVR plateau values varied from 19 ± 3 (F_15_) to 42,207 ± 1848 Pa (F_10_) (Table 6).

In addition to the yield point (*τ*_0_), the flow point (*τ_f_*) is likewise an important CQA, representing the shear stress value where the moduli crossover (*G’*= *G″*). *τ_f_* is considered the borderline between the gel/solid and the fluid/liquid-like states. When overcoming such points, any disturbance at the microstructure produces irreversible deformations [60]. *τ*_0_ and *τ_f_* responses ranged from 3.6 ± 0.6 Pa (F_15_) to 74 ± 10 Pa (F_10_) and from 3.6 ± 1 (F_15_) to 104 ± 15 Pa (F_10_), respectively (Table 6).

The storage modulus (*G*) represents the magnitude of energy stored in a material, whereas the loss modulus (*G″*) represents the energy loss due to viscous dissipation. Therefore, a material will present elastic properties at *G’* > *G″* and viscous properties at *G’* < *G″* [73]. During a deformation process, the prevalence of elastic properties determines a more stable structure, since reversible deformations (*G’*) overlap the irreversible ones (*G″*). *G’* and *G″* results were found within the range of 84 ± 5 (F_1_) to 50,732 ± 1381 Pa (F_10_) and 29.7 ± 1.3 (F_1_) to 18,732 ± 1150 Pa (F_10_), respectively (Table 6). At 1 Pa, *G’* and *G″* values of the overall DoE formulations were found within the LVR plateau, indicating the suitability of this shear stress to be used in frequency sweep tests.

Important considerations are likewise extracted from the loss tangent (*tan δ*) response, a dimensionless term that describes the ratio between the *G″* and the *G’*. This CQA may be usefully employed to elicit information regarding system structure. With a *tan δ* < 1 (*G″* < *G’*), the elastic structure predominates, presenting a gel-like or solid state, whereas, if the *tan δ* > 1 (*G″* > *G’*), the system presents more viscous properties and a liquid or fluid state, and when the *tan δ* = 1 (*G″* = *G’*), the *τ_f_* is achieved [60]. The *ta**n δ* response varied from 0.231 ± 0.005 (F_8_) to 0.513 ± 0.016 (F_15_) (Table 6), which is suitable to support a topical application.

At different factor level combinations, significant differences were observed for the studied CQAs (*p* < 0.05).

As displayed in Appendix A, glycerol monostearate concentration (x_1_) was the most impactful factor, exerting a positive effect (Prob > |t| < 0.05) on the CQAs. *τ*_0_ was the only response significantly influenced by isopropyl myristate amount (x_2_), but in an antagonistic manner.

Concerning the viscoelastic results, a direct relationship among those CQAs was observed. Therefore, at high x_1_ levels, the LVR plateau, *τ*_0_ and *τ_f_* significantly increased.

As previously mentioned, the thickener agent had a synergetic contribution to the droplet stability and gel network structure. Greater amounts of glycerol monostearate produce larger droplet sizes and more viscous systems, which reinforce van der Waals and electrostatic forces. Viscoelastic responses are highly dependent on the strength of these attractive and repulsive forces and thus on the formulation structure. Therefore, the stronger the intermolecular interactions, the greater the LVR plateau, *τ*_0_ and *τ_f_* values, since a higher network structure strength offers more deformation resistance to external forces, requiring higher shear values to initiate flow (*τ*_0_) and even structural breakdown (*τ_f_*). As such, those CQAs are important stability indicators [58,74,75]. This trend is consistent with the *τ*_0_ values extracted from Herschel-Bulkley, Bingham and Casson flow models (Table 7).

At medium x_1_ level, DoE formulations exhibited lower *τ*_0_ and *τ_f_* values, suggesting that a small shear was needed to initiate flow, which may ascribe the better spreadability of the formulation to the skin [76]. 

In contrast, at high x_2_ levels, formulations are more prone to deformation forces. In the presence of superior amounts of isopropyl myristate, a lower LVR plateau, *τ*_0_ and *τ_f_* are obtained, as a result of the reduction in the oily phase melting temperature along with the loss in system viscosity. 

The viscoelastic moduli are also measurements of molecular interaction, reflecting the structural characteristics of the cream formulations. DoE formulations exhibited a prevalence of elastic properties with *G’* (the solid-like component) significantly higher than *G″* (the fluid-like component), suggesting the presence of a consistent gel network structure dominated by cohesive forces. Systems with such behavior (*G’* > *G″*) present lesser separation phenomena and higher resistance to deformation forces [3,77]. Like other viscoelastic parameters, *G’* and *G″* are also pointed out as important stability indicators.

At a high x_1_ level, a rise in *G’* and *G″* values was observed. The acquired results highlight the contribution of the thickener to forming a more structured gel network and solid-like formulation properties [69]. This is in agreement with the *τ*_0_ values, required to initiate flow, and the longer time for the structural recovery, as shown by S_R_ values.

When increasing the amount of glycerol monostearate, an inverse trend was observed for *tan (δ)* with values less than 0.52, also confirming the prevalence of elastic behavior. 

#### 4.3.4. Product Performance

##### IVRT

Regarding the impact of formulation and process variables on cream release profile, the cumulative % of HC released ranged from 5.1 ± 0.4 (F_8_) to 12.4 ± 0.2 % (F_15_) after 6 h (R_6h_) and from 9.7 ± 0.8 (F_8_) to 25.3 ± 1.2% (F_15_) after 24 h (R_24h_) (Table 8). As presented in Figure 4, release profiles evidence a biphasic pattern. Initially, the release rate is more pronounced; however, after 10 h, it becomes slower. At different factor level combinations, significant differences were observed for the considered CQAs (*p* < 0.05).

Release kinetics extracted from mathematical fitting enable a preliminary indication of drug release mechanisms from the vehicle, in the absence of the foremost biological barrier, the SC [78]. Accordingly, it is seen that all the formulations follow a Higuchi diffusion model (c_1_.√t), with the release rate (c_1_) changing from 112 ± 2 (F_6_) to 196 ± 7 μg/cm^2^/√t (F_15_). These results are in accordance with expected data for corticoid semisolid formulations [79]. Higuchi’s model proves that the active substance is gradually transferred from the vehicle through a linear concentration gradient. As the dispersed phase behaves like a drug reservoir, a mass transfer is observed across the emulsion phases and towards the membrane, prolonging HC release [80]. This linear relationship was attained for all DoE formulations with R^2^ values superior to 0.96899.

The Korsmeyer-Peppas model (*k*t^c2^) model allows us to characterize the different release mechanisms through the evaluation of the diffusion release exponent (c_2_). Four scenarios may be possible: (i) c_2_ close to 0.5—Fickian diffusion process, and non-Fickian diffusion process where (ii) 0.5 < c_2_ < 1.0—anomalous transport, (iii) c_2_ = 1.0—zero-order model, and finally (iv) c_2_ >1.0—super case-II transport [81]. Taking into account this classification, it is seen that DoE formulations displayed a hybrid behavior between Fickian and anomalous (non-Fickian) transport [0.447 ± 0.024 (F_4_) < c_2_ < 0.629 ± 0.019 (F_12_)], ascribed to the differences in the microstructure network (Table 8).

As shown in Appendix A, glycerol monostearate amount (x_1_) and homogenization rate (x_3_) interaction were found to have an important antagonistic effect on R_24h_ response (Prob > |t| < 0.05) (Figure 5).

Taking into account independent variables’ impact, low levels of x_1_ and high levels of x_3_ result in greater c_1_ and R_24h_, probably due to their contribution to the lower formulation viscosity and smaller droplet size.

According to the Stokes law, the diffusion coefficient of the active substance is inversely proportional to matrix viscosity, since more viscous formulations will retain the active substance, hindering its release from the vehicle for longer [82,83]. In turn, by decreasing formulation viscosity, a superior molecular mobility will be produced, leading to greater release rate [84]. Pertaining to DoE formulations, this trend was observed for a low viscous formulation (F_15_) presenting the best c_1_, as well as, R_24h_ results. However, more viscous systems also presented elevated results for drug release [85]. This unexpected behavior can be ascribed to smaller droplet sizes and particular rheological aspects, such as a thixotropic behavior and viscoelastic properties.

DoE formulations with small droplets had a superior c_1_ and R_24h_ values due to the increase in the total surface area [86]. Formulations with higher S_R_ values demonstrated an enhancement of the release profile, since longer recovery periods result in superior HC release times [7]. Greater values of c_1_ and R_24h_ were also observed for formulations presenting rising values of *G’*. Formulation retention and contact time on the skin surface are governed by the viscoelastic properties. Therefore, to improve cream retention at the application site and to offer a prolonged controlled release platform, this formulation should be predominantly elastic, exhibiting high *G’* values [87].

In turn, during cream manufacturing, homogenization is a critical stage, greatly influencing drug dissolution and, consequently, product homogeneity. A homogenization rate of 22,000 rpm negatively impacted drug content distribution. Hence, unexpected release rate values could be a result of homogeneity loss.

Although not presenting a significant impact in the different fitted models, isopropyl myristate amount (x_2_) is an important variable in terms of product performance because its solvent/enhancer function contributes to the enhancement of drug bioavailability. As HC transport through the vehicle follows a diffusion mechanism, the main release driving force is the concentration gradient of the dissolved HC. Therefore, at high levels of x_2_, HC release from the vehicle will be favored [88].

##### IVPT

Assessing the impact of formulation and process variables on cream permeation profile, the cumulative amount of HC that permeated across the split skin varied from 1.6 ± 0.9 (F_1_ and F_13_) to 1.9 ± 1.1 μg/cm^2^ (F_4_ and F_14_) after 6 h (Q_6h_), from 3.0 ± 1.7 (F_6_) to 9 ± 5 μg/cm^2^ (F_11_) after 24 h (Q_24h_) and from 11 ± 6 (F_6_ and F_15_) to 40 ± 23 μg/cm^2^ (F_11_) after 48 h (Q_48h_) (Table 10). The flux (J_ss_), determined from the slope of the resulting linear plot region, which varied from 0.16 ± 0.03 (F_12_) to 0.97 ± 0.08 μg/cm^2^/h (F_11_). K_p_ was found to range between 0.542 × 10e^−02^ (F_12_) and 2.55 × 10e^−02^ cm/h (F_11_) (Table 9).

As evident in Figure 6, the permeation profiles also suggest a biphasic pattern, characterized by a slow permeation up to the first 10 h, followed by an increase in HC permeation amount up to 48 h. It is interesting to observe that DoE formulations exceeded permeation expectations, showing better results than those presented in QTPP specifications.

At different factor level combinations, significant differences in J_ss_, Q_24h_ and Q_48h_ were observed (*p* < 0.05).

When performing a comparative analysis of in vitro responses, it is interesting to observe that formulations with higher release performance do not present the best permeation responses. In turn, DoE formulations with low c_1_ and R_24h_ values show an improvement in their permeation performance. Hence, a direct relationship between HC release and permeation may not be established since the permeation process considers the limiting SC barrier.

Apart from the non-significance of the factors, considering the coefficient signal and magnitude displayed in Appendix A, glycerol monostearate amount (x_1_) provides a positive impact, while isopropyl myristate amount (x_2_) and homogenization rate (x_3_) induced an antagonistic effect on permeation responses.

As aforementioned, glycerol monostearate concentration produces major changes for formulation viscosity. Hence, at high levels of x_1_, a general trend towards HC retention into more viscous systems is expected, limiting its skin uptake and diffusion. Indeed, viscosity has long been described as an important factor for semisolid formulations as it may influence the release of drug by generally limiting the diffusion rate from the vehicles and, consequently, the drug available for skin permeation [89,90]. Conversely, the acquired results demonstrate that medium levels of x_1_ are preferred for the achievement of greater penetration results. The results also suggest that the active substance release from more viscous systems is not a limiting stage for its skin uptake. A suitable HC bioavailability was ensured locally.

High levels of x_1_ also contribute to superior emollient and occlusive effects. The occlusion leads to an increase in skin hydration by the swelling and softening of the SC structure, improving drug penetration through the skin [91,92,93]. Furthermore, previous experiments also demonstrate a significant impact of the x_1_ variable on the cream’s mechanical properties. The higher amount of glycerol monostearate significantly impacts the adhesive properties [1]. A prolonged contact between the formulation and skin contributes to enhanced permeation responses.

Considering homogenization rate (x_3_), a reduced droplet size was attained at 22,000 rpm. Conversely to the IVRT results, smaller droplets yielded low permeation parameter values.

Isopropyl myristate amount (x_2_) seems to present an important role in HC permeation behavior. According to diffusion Fick’s first law, the increase in J_ss_ directly depends on the drug concentration in the vehicle (C_0_) and on the drug permeability coefficient (K_p_). The latter is given by the product of the drug partition coefficient between SC and the vehicle (P) by the diffusion coefficient (D), divided by the diffusion path [8,94]. In DoE formulations, C_0_ was kept constant (1%). Hence, superior values of K_p_ and J_ss_ may be attributed to the penetration enhancer contribution to increase HC D and P values. As a permeation enhancer, isopropyl myristate interacts and fluidizes the rigid intercellular SC lipid bilayers, changing their solubility. Through this mechanism, high levels of x_2_ favor the HC diffusion coefficient (D) and its partition coefficient (P) to the skin, enhancing drug permeation in a synergetic manner [95,96,97].

Deviations in permeation responses could also be attributed to HC lipophilicity (log P = 1.61, Biopharmaceutics Classification System Class II), hindering its diffusion through the different skin layers, particularly from the outer skin layer to the more aqueous environments (epidermis tissue) [29,98]. However, this observation is not a real drawback, since it addresses a safer topical administration.

Considering that HC is enough solubilized in the formulation to ensure vehicle-skin interface saturation, the greatest permeation parameters result from the vehicle-skin interactions, rather than HC-skin interactions [63].

#### 4.3.5. Stability Protocol

##### Assay

When inspecting the impact of formulation and process variations on cream stability, the drug assay ranged from 81.9% (F_15_) to 120.5% (F_13_) (Table 5). Notwithstanding the non-significance of the model, as represented in Appendix A, the x_1_x_3_ interaction term was found to have an important synergistic effect (Prob > |t| < 0.05) on the active substance assay (Figure 7). 

According to data analysis, a balance between the thickening agent (_x1_) and the mechanical energy (x_3_) applied to the system must be established since, when the concentration of the glycerol monostearate is increased, there is a rise in dispersed phase viscosity and, consequently, in the viscosity of the whole system. Thereby, during more viscous systems manufacturing, a superior homogenization rate must be applied for the achievement of a more efficient homogenization process and formulation homogeneity.

Previous experiments show that formulation and process variability display a significant impact on drug content distribution, with separation mechanisms pointed out as the main reason for assay variations [9,99,100].

##### pH

Considering the effect of formulation and process variability on cream stability, significant variations were not detected in pH (6.570 ± 0.008 (F_6_)–6.75 ± 0.02 (F_12_) (Table 5). At different factor level combinations, pH results remained relatively constant (*p* < 0.05). 

As presented in Appendix A, there is no important effect on pH response with x_1_, x_2_ and x_3_ variables (Prob >|t| > 0.05).

Note that the acquired results are slightly above the skin physiological pH range (5–6.5). It is stated that topical formulations pH between 5 and 7 seem not to cause skin irritation, which denotes a safe application of the DoE formulations [101]. This pH range also ensures preservatives’ effectiveness (pH 4–8) and HC solubility and permeation (pka = 12.59) [1,102].

##### Instability Index, Sedimentation and Creaming Rate

When assessing the impact of formulation and process variability on cream stability, the instability index varied from 0.031 ± 0.012 (F_10_) to 0.28 ± 0.13 (F_3_). Moreover, the sedimentation rate ranged from 0.031 ± 0.002 (F_15_) to 0.255 ± 0.124 μm/s (F_1_) and the creaming rate from 0.009 ± 0.000 (F_14_) to 0.38 ± 0.07 μm/s (F_15_) (Table 5). At different factor level combinations, we also observed significant differences in creaming rate (*p* < 0.05).

As shown in Appendix A, glycerol monostearate amount (x_1_) and isopropyl myristate (x_2_) produced opposite effects on the instability index and creaming rate (Prob >|t| < 0.05).

At high levels of x_1_, lower values of instability index and creaming velocity were attained. Such behavior is ascribed to the thickening effect in the dispersed phase and formulation viscosity, and to viscoelastic properties. More viscous systems entail a reduction in droplet movement and lesser aggregation/coalescence events. These results are in accordance with Stokes law assumptions. The higher the formulation viscosity, the better the physical stability [103]. Furthermore, formulations with a wide LVR plateau and *τ*_0_ present high system rigidity and thus an exceptional stability against separation phenomena [104].

In contrast, at high levels of isopropyl myristate (x_2_), the reduced system viscosity prompts an increased instability index and creaming velocity. Moreover, a higher amount of isopropyl myristate results in smaller droplets. Generally, formulations with larger droplet sizes show higher instability index values, while those with smaller globules exhibit more resistance to instability phenomena.

Although not statistically significant, a separation phenomenon was detected during stability analysis and we found an antagonistic effect of the homogenization rate (x_3_) on the considered CQA. High levels of x_3_ result in smaller droplet sizes and superior viscosity, preventing globule movements and, eventually, separation mechanisms.

It is possible to observe that the prevalence of sedimentation or the creaming process in emulsion-based formulations relied on their droplet size. DoE formulations with larger globules demonstrated an upper incidence of the sedimentation process, while formulations with smaller ones exhibited a prevalent creaming process.

The physical phenomena involved in each breakdown process are not simple and require a thorough analysis of the involved surface forces. Under centrifugal forces, severe variations in transmission profiles comprise information about the breakdown behavior of the individual samples. When such forces exceed the Brownian motion (erratic motion of the oil droplets, arising from their random collisions), a concentration gradient arises in the system, with the larger droplets moving faster to the bottom (if their density is larger than that of the medium) or to the top (if their density is lower than that of the medium) of the cell, disclosing sedimentation and/or creaming mechanisms, respectively [75,103,105]. Such outcomes are directly related to the physical stability of the emulsion-based products: the lower the sedimentation and creaming velocity, the higher the cream stability.

Taking into account previous experiments, it was inferred that formulation viscosity has supremacy over droplet size, with the most viscous formulations showing greater globule sizes and a lower separation velocity. Therefore, these results corroborate the importance of viscosity as stability indicator.

#### 4.3.6. Overall Outlook

The outcome of this study was to establish a correlation between formulation and process variability, product microstructure and performance. As per the results, such an approach is not a straightforward and well-established procedure, requiring the assembly of different synergistic and antagonistic effects. A summary of the tripartite analysis is available in Table 10.

#### 4.3.7. Summing-Up

♦Glycerol monostearate amount is a critical material attribute, due to its significant impact on formulation droplet size, rheological properties, physical stability and IVRT results.♦An important contribution to drug-vehicle-skin interaction is given by glycerol monostearate.♦Isopropyl myristate amount presents a wide impact on formulation physical stability.♦As permeation enhancer, isopropyl myristate plays an important role in drug penetration through the skin. ♦Formulation droplet size and, consequently, physical stability are highly dependent on homogenization rate. ♦Glycerol monostearate amount and homogenization rate interaction demonstrated to govern HC release.♦Isopropyl myristate and homogenization rate interaction seems to significantly influence formulation droplet size.

### 4.4. Optimal Working Conditions

Design space (DS) is a multidimensional combination and interaction between the independent variables that provide assurance of quality. In DS, optimal CMAS and CPPs working ranges are established, within which QTPP specifications can be achieved. Working within the design space is not considered a change, since different experimental conditions may produce the same qualified product [17,106].

As represented in Figure 8, an optimal region was established by overlaying the contour plots of the overall CQAs. Three separate optimal regions, (a), (b) and (c), were acquired. The acceptance criteria for the establishment of optimal regions are listed in Table 3.

The overlay of a, b and c white regions represents the optimal experimental conditions, where every single point corresponds to a combination of two variables, resulting in a singular formulation within the predefined acceptable limits [107,108].

As evidenced, x_1_ and x_2_ are very close to the medium level, while x_3_ is slightly closer to the high level. Therefore, a glycerol monostearate content of 10%, an isopropyl myristate content of 5.5%, and a homogenization rate around 20,000 rpm will ensure a more robust and flexible process, which invariably meets the required QTPP specifications.

## 5. Conclusions

Designing cream formulations is not a trivial task due to their complex nature and the lack of knowledge of the interconnections between the material and/or process variables.

In this work, a QbD approach was successfully applied to an o/w cream optimization. FMECA proved to be a helpful risk analysis tool, enabling the identification and the ranking of the most critical factors. Box-Behnken design was applied to improve the fundamental understanding of CMAs and CPP effects, and their interactions, on product quality profile, more precisely on cream microstructure, performance and stability. From a combinatorial factor analysis, the glycerol monostearate amount (x_1_), followed by the homogenization rate and (x_3_), were identified as the important factors for droplet size, rheological properties, assay instability index and creaming rate response. In IVRT responses, x_1_ and x_3_ variables demonstrated an important impact on the active substance release from the vehicle due to their significant effect on microstructural features, while IVPT responses were majorly impacted by the x_2_ variable ascribed to their fundamental interaction with the biological membrane, contributing to a more effective permeation. Moreover, the in vitro methodologies revealed their great ability to discriminate product variability.

A design space that meets the predefined QTPP specifications was ultimately established, specifying the optimal operating ranges for the most relevant variables, within which product variability is certainly minimized.

From the above findings, it can be concluded that, as an optimization instrument, the QbD approach presents significant benefits to the pharmaceutical industry, since a detailed understanding of cream formulation and process parameters may reduce product variability, ensuring its final quality, time- and cost-saving procedures and regulatory flexibility.

## Figures and Tables

**Figure 1 pharmaceutics-12-00647-f001:**
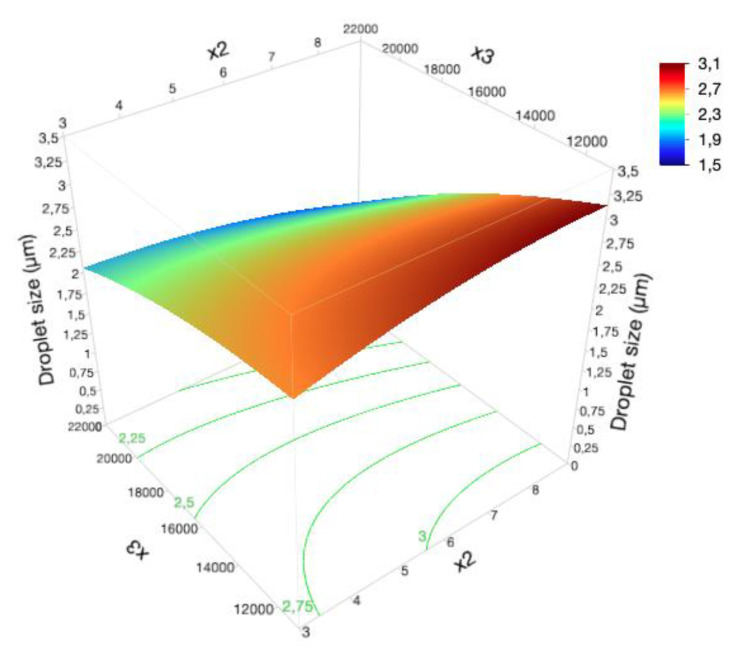
Response surface plot showing the effect of isopropyl myristate amount and homogenization rate interaction on droplet size response.

**Figure 2 pharmaceutics-12-00647-f002:**
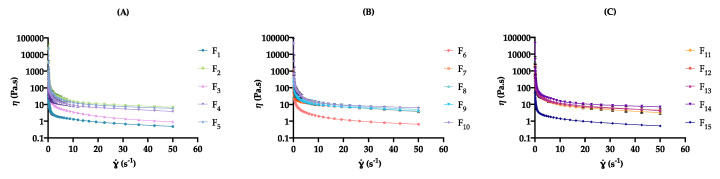
Effect of independent variables on cream viscosity: (**A**) F_1_–F_5_, (**B**), F_6_–F_10_ and (**C**) F_11_–F_15_.

**Figure 3 pharmaceutics-12-00647-f003:**
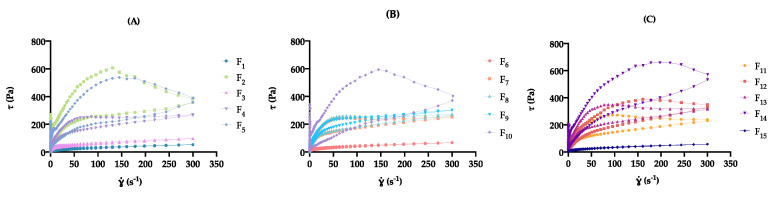
Effect of independent variables on cream thixotropic behavior: (**A**) F_1_–F_5_, (**B**), F_6_–F_10_ and (**C**) F_11_–F_15_.

**Figure 4 pharmaceutics-12-00647-f004:**
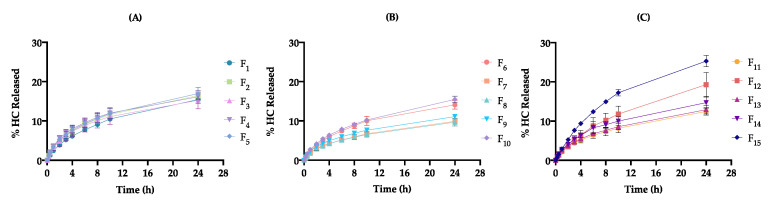
Effect of independent variables on cream release profile: (**A**) F_1_–F_5_, (**B**), F_6_–F_10_ and (**C**) F_11_–F_15_.

**Figure 5 pharmaceutics-12-00647-f005:**
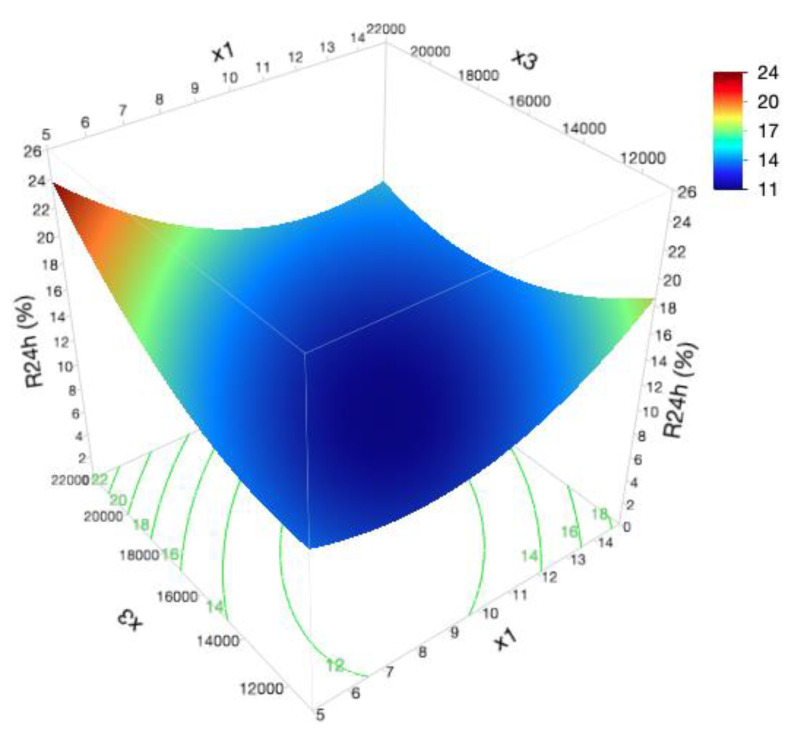
Response surface plot showing the effect of glycerol monostearate amount and homogenization rate interaction on R_24h_ response.

**Figure 6 pharmaceutics-12-00647-f006:**
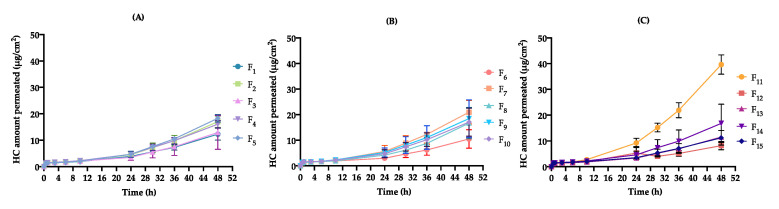
Effect of independent variables on cream permeation profile: (**A**) F_1_–F_5_, (**B**), F_6_–F_10_ and (**C**) F_11_–F_15_.

**Figure 7 pharmaceutics-12-00647-f007:**
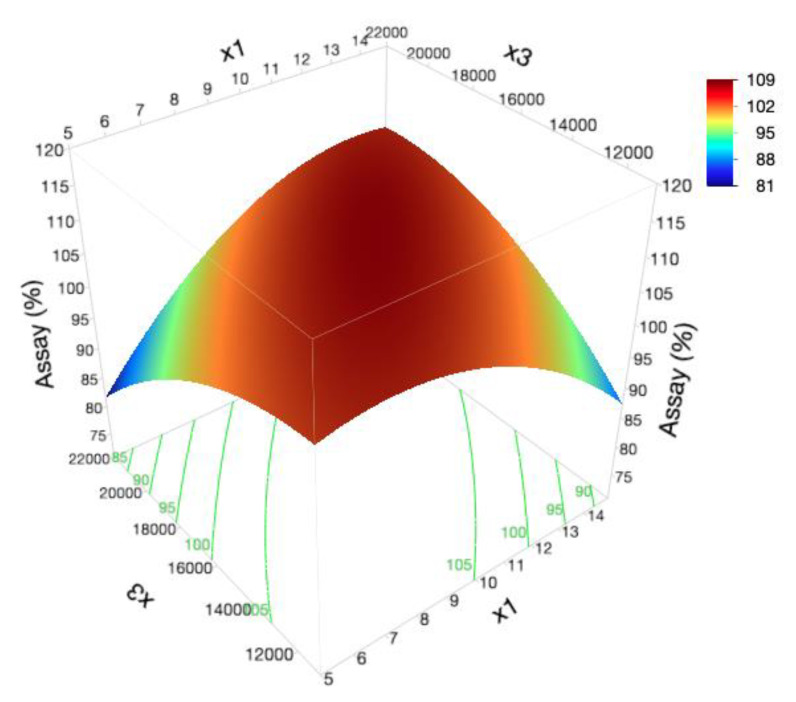
Response surface plot showing the effect of glycerol monostearate amount and homogenization rate interaction on assay response.

**Figure 8 pharmaceutics-12-00647-f008:**
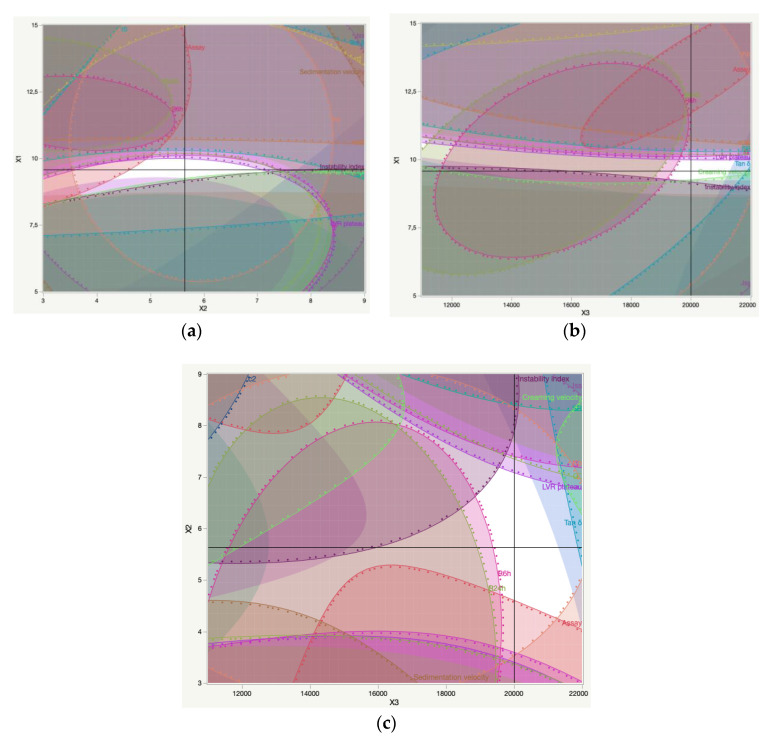
Design space of glycerol monostearate content, isopropyl myristate content and homogenization rate, comprising the overlay of (**a–c**) optimal regions.

**Table 1 pharmaceutics-12-00647-t001:** Coded values of independent experimental variables.

Independent Variables	Level		
	−1	0	+1
x_1_: Glycerol monostearate amount (%)	5	10	15
x_2_: Isopropyl myristate amount (%)	3	6	9
x_3_: Homogenization rate (rpm)	11,000	16,000	22,000

**Table 2 pharmaceutics-12-00647-t002:** Experimental planning according to Box-Behnken design.

ID	x_1_ (%)	x_2_ (%)	x_3_ (rpm)
F_1_	5	3	16,000
F_2_	15	9	16,000
F_3_	5	6	11,000
F_4_	10	9	11,000
F_5_	15	6	11,000
F_6_	5	9	16,000
F_7_	10	6	16,000
F_8_	10	6	16,000
F_9_	10	3	11,000
F_10_	15	3	16,000
F_11_	10	6	16,000
F_12_	10	9	22,000
F_13_	10	3	22,000
F_14_	15	6	22,000
F_15_	5	6	22,000

**Table 3 pharmaceutics-12-00647-t003:** Quality target product profile (QTPP) specifications and critical quality attribute (CQA) identification of a hydrocortisone cream formulation.

Drug Product Quality Attributes	Target	Is it a CQA?	Justification
*Dosage form*	Cream	-	Emulsion-based semisolid product assists in topical delivery improvement.
*Route of administration*	Topical	-	Local administration avoids systemic side effects. Non-invasive, convenient and painless administration. High patient compliance.
*Dosage strength*	1 % *w*/*w*	-	1 % hydrocortisone ensures formulation efficacy.
*Dosage form design*	o/w emulsion with solubilized hydrocortisone	-	Biphasic semisolid systems are vehicles that enable an appropriate delivery of hydrocortisone to the target skin layer.
***Assay***	90.0–110.0 % of the labelled claim; RSD NMT 6.0%	Yes	Influence on therapeutic efficacy.
*Physical attributes*			
Appearance	White smooth cream	No	Not directly related with safety and efficacy.
Color	No addition of artificial colors	No	Required to ensure patient compliance and acceptance.
Odor	No objectionable odor	No	Impact on physical and chemical stability.
**pH**	5.5–7.0	Yes	Compatible with skin pH to prevent local irritation.
**Droplet size**	2.0–4.5 μm	Yes	Impact on drug product efficacy and stability.
Rheological aspects			
*η* _10_	6.0–8.0 Pa.s	Yes	Impact on cream spreadability which is important for patient compliance. Influence on in situ cream persistence and consequently its duration of action. Influence on physical stability. Impact on drug release and diffusion rate at the microstructure level.
**Rheological behavior**	Non-Newtonian, pseudoplastic pattern	Yes	
**Rheological model**	Herschel-Bulkley and Cross	Yes	
**S_R_**	10,000–20,000 Pa/s	Yes	
**LVR plateau**	3000–5000 Pa	Yes	
***τ*_0_**	35.0–50.0 Pa	Yes	
***τ_f_***	55.0–65.0 Pa	Yes	
***G’***	4500–5500 Pa	Yes	
***G″***	1500–2000 Pa	Yes	
***tan δ***	0.35	Yes	
*Product performance*			
IVRT			
**c_1_**	>120–125 μg/cm^2^/√t	Yes	To ensure therapeutic efficacy. Useful to assess the sameness of the dosage form. Reflect the effect of formulation and/or process parameters on cream microstructure.
**k**	>2.5–3 t^−1^	Yes ^a^	
**c_2_**	>0.45–0.55	Yes	
**R_6h_**	>6.35–10.0%	Yes	
**R_24h_**	>12.25–20%	Yes	
**IVPT**			
**J_ss_**	>0.25–0.35 μg/cm^2^/h	Yes	Impact on therapeutic efficacy. Critical to detect particular differences regarding the hydrocortisone permeation rate and extent through the skin. Important to better understand the impact of formulation and/or process parameters.
**ER**	1	Yes ^a^	
**k_p_**	>1.06 × 10^−2^ cm/h	Yes	
**Q_6h_**	>0.8–3.0 μg/cm^2^	Yes	
**Q_24h_**	>2.0–8.0 μg/cm^2^	Yes	
**Q_48h_**	>6.0–15.0 μg/cm^2^	Yes	
t_lag_	10 h	Yes ^a^	
*Physical stability*			
**Instability index**	NMT 0.13	Yes	Critical to forecast physical stability. Important to maintain formulations performance during the storage period.
**Sedimentation rate**	NMT 0.15 μm/s	Yes	
**Creaming rate**	NMT 0.08 μm/s	Yes	

Key: European Pharmacopeia (Eur.Ph.); not more than (NMT); oil-in-water (o/w); relative standard deviation (RSD); United States Pharmacopeia (USP); ^a^ Formulation and process variables will have no impact upon this CQA, but it is considered as a QTPP element. The investigated CQAs are highlighted (in bold).

**Table 4 pharmaceutics-12-00647-t004:** Failure Mode, Effects and Criticality Analysis (FMECA) tool presenting initial risk assessment for cream formulation.

Category	Risk Area	Variables	Failure Mode	Failure Cause	Failure Effect	S	P	D	RPN
CMAs	Formulation	API	Inadequate phase solubilization Low/excessive concentration		Non-homogeneity.	5	2	3	30
		Emollients		Weighing error Lack of scientific knowledge Lack of detail formulation understanding Lack of excipients function	Cream with inappropriate structure-form.	3	2	1	6
		Emulsifying agent			Undesirable droplet size. Physical instability.	5	3	1	15
		Stiffening agent			Inadequate rheological properties. Inadequate drug release and permeation. Physical instability.	4	3	4	48
		Permeation enhancer			Inadequate drug permeation.	5	3	3	45
		Alkalizing agent			Skin irritancy. Inadequate rheological properties. Inadequate drug release and permeation. Physical instability. Chemical instability.	5	2	1	10
		Humectant			Cream with inappropriate structure-form.	3	1	1	3
		Antioxidants			Chemical instability.	5	2	1	10
		Preservatives			Microbiological instability.	5	2	1	10
		Solvent			Non-homogeneity. Drug recrystallization.	5	2	1	10
		Purified water			Cream with inappropriate structure-form.	5	2	1	10
CPPs	Production process	Equipment type	Inappropriate shear mechanism Low/excessive bend/homogenization time/rate Low/excessive blend/homogenization temperature Equipment stop inadvertently	Lack of process monitoring Lack of scientific knowledge Lack of equipment specifications knowledge Malfunction of the equipment	Non-homogeneity. Undesirable droplet size. Physical instability.	5	3	1	15
		Rotor–stator rod			Non-homogeneity. Physical instability.	5	3	1	15
		De-aeration via vacuum			Excessive air entrapment.	3	2	2	12
		Phase addition order			Undesirable droplet size. Physical instability.	5	2	1	10
		Blending temperature			Non-homogeneity. Impurities. Chemical instability. Premature drug crystallization.	4	3	1	12
		Blending rate			Non-homogeneity. Undesirable droplet size. Physical instability.	5	2	1	10
		Blending time			Non-homogeneity. Undesirable droplet size. Physical instability.	5	3	1	15
		Homogenization temperature			Non-homogeneity. Impurities. Chemical instability. Premature crystallization.	5	3	1	15
		Homogenization rate			Non-homogeneity. Undesirable droplet size. Inadequate rheological properties. Inadequate drug release and permeation rate. Physical instability.	5	4	2	40
		Homogenization time			Non-homogeneity. Undesirable droplet size. Inadequate rheological properties. Inadequate drug release and permeation rate. Physical instability.	5	3	1	15
		Cooling rate			Non-homogeneity. Inadequate rheological properties. Inadequate drug release and permeation rate. Physical instability.	3	3	1	9

Key: Active ingredient substance (API); severity (S); probability of occurrence (P); detectability (D).

**Table 5 pharmaceutics-12-00647-t005:** Effect of independent variables on different cream CQAs.

ID	Droplet Size (μm)	Assay (%/RSD)	pH	Instability Index	Sedimentation Rate (μm/s)	Creaming Rate (μm/s)
F_1_	2.49 ± 0.86 C	92.0/0.8 C	6.65 ± 0.04 C	0.270 ± 0.093 NC	0.255 ± 0.124 NC	0.24 ± 0.04 NC
F_2_	2.74 ± 0.81 C	93.0/ 1.1 C	6.73 ± 0.04 C	0.077 ± 0.006 C	0.15 ± 0.05 NC	0.03 ± 0.03 C
F_3_	3.1 ± 1.0 C	101.6/4.1 C	6.60 ± 0.05 C	0.28 ± 0.13 NC	0.08 ± 0.03 C	0.37 ± 0.08 NC
F_4_	3.0 ± 1.0 C	106.7/9.9 NC	6.62 ± 0.06 C	0.17 ± 0.05 NC	0.11 ± 0.03 C	0.099 ± 0.006 NC
F_5_	3.2 ± 1.0 C	87.1/0.1 NC	6.653 ± 0.012 C	0.072 ± 0.006 C	0.19 ± 0.07 NC	0.019 ± 0.011 C
F_6_	2.26 ± 0.50 C	118.5/15.5 NC	6.570 ± 0.008 C	0.3 ± 0.2 NC	0.12 ± 0.02 C	0.36 ± 0.13 NC
F_7_	2.70 ± 0.68 C	111.9/0.9 NC	6.677 ± 0.012 C	0.116 ± 0.007 C	0.13 ± 0.02 C	0.05 ± 0.02 C
F_8_	2.72 ± 0.68 C	107.3/0.2 C	6.660 ± 0.009 C	0.113 ± 0.003 C	0.10 ± 0.03 C	0.05 ± 0.02 C
F_9_	2.6 ± 1.0 C	112.6/0.6 NC	6.680 ± 0.008 C	0.071 ± 0.004 C	0.19 ± 0.08 NC	0.03 ± 0.02 C
F_10_	2.59 ± 0.79 C	110.7/0.4 NC	6.675 ± 0.012 C	0.031 ± 0.012 C	0.11 ± 0.04 NC	-
F_11_	2.50 ± 0.77 C	108.7/1.1 C	6.733 ± 0.017 C	0.14 ± 0.02 NC	0.11 ± 0.03 C	0.06 ± 0.01C
F_12_	1.63 ± 0.36 C	84.4/3.1 NC	6.75 ± 0.02 C	0.118 ± 0.004 C	0.12 ± 0.03 C	0.05 ± 0.02C
F_13_	2.15 ± 0.62 C	120.5/ 0.8 NC	6.673 ± 0.005 C	0.06 ± 0.02 C	0.052 ± 0.000 C	0.03 ± 0.02 C
F_14_	2.17 ± 0.69 C	112.0/1.7 NC	6.74 ± 0.00 C	0.048 ± 0.003 C	0.082 ± 0.009 C	0.009 ± 0.000 C
F_15_	1.40 ± 0.28 C	81.9/ 2.1 NC	6.627 ± 0.005 C	0.27 ± 0.12 NC	0.031 ± 0.002 C	0.38 ± 0.07 NC

Key: Compliant (C); noncompliant (NC); relative standard deviation (RSD). Rheological properties. Rotational measurements.

**Table 6 pharmaceutics-12-00647-t006:** Effect of independent variables on cream rheological profile. Data are expressed as mean ± SD (*n* = 3).

ID	Rotational Measurements		Oscillatory Measurements					
	CR Step Test	CR Ramp Test	Amplitude Sweep Test			Frequency Sweep Test at 1 Hz		
	*η*_10_ (Pa.s)	*SR* (Pa/s)	LVR (Pa) plateau	*τ*_0_ (Pa)	*τf* (Pa)	*G’* (Pa)	*G″* (Pa)	*tan δ*
F_1_	0.91 ± 0.11 NC	942 ± 224 NC	52 ± 3 NC	6.1 ± 0.4 NC	7.4 ± 1.1 NC	84 ± 5 NC	29.7 ± 1.3 NC	0.356 ± 0.008 NC
F_2_	11.1 ± 0.9 NC	71,770 ± 3532 NC	26,613 ± 458 NC	40 ± 5 NC	102 ± 14 NC	29,621 ± 223 NC	9846 ± 250 NC	0.332 ± 0.011 C
F_3_	2.08 ± 0.02 NC	2225 ± 125 NC	330 ± 38 NC	5.6 ± 0.7 NC	10.0 ± 0.5 NC	337 ± 15 NC	86.24 ± 0.91 NC	0.256 ± 0.009 C
F_4_	6.5 ± 0.6 NC	13,417 ± 533 C	3257 ± 335 C	10.4 ± 0.6 NC	13.4 ± 1.3	3242 ± 265 NC	916 ± 9 NC	0.28 ± 0.02 C
F_5_	8.5 ± 1.0 NC	62,847 ± 7888 NC	28,510 ± 623	32 ± 3 NC	64 ± 2 C	30,522 ± 858 NC	11,247 ± 151 NC	0.369 ± 0.007 NC
F_6_	1.265 ± 0.007 NC	1157 ± 24 NC	106 ± 3 NC	5.63 ± 0.02 NC	7.90 ± 0.08 NC	130 ± 6 NC	34.0 ± 0.5 NC	0.262 ± 0.011 C
F_7_	7.2 ± 0.6 C	12,720 ± 401 C	3226 ± 262 C	13 ± 3 NC	21 ± 2 NC	3465 ± 106 NC	855 ± 20 NC	0.247 ± 0.003 C
F_8_	7.7 ± 0.3 C	14,600 ± 640 C	3549 ± 328 C	28 ± 5 NC	35 ± 9 NC	3374 ± 236 NC	779 ± 72 NC	0.231 ± 0.005 C
F_9_	7.1 ± 0.2 C	6319 ± 310 NC	2907 ± 59 NC	51 ± 2 NC	69 ± 2 NC	3191 ± 173 NC	840 ± 22 NC	0.264 ± 0.008 C
F_10_	9.5 ± 0.3 NC	78,470 ± 3401 NC	42,207 ± 1848 NC	74 ± 10 NC	104 ± 15 NC	50,732 ± 1381 NC	18,732 ± 1150 NC	0.369 ± 0.013 NC
F_11_	6.4 ± 0.8 NC	21,017 ± 927 NC	6259 ± 526 NC	12.8 ± 1.4 NC	20 ± 3 NC	6159 ± 351 NC	1878 ± 92 C	0.305 ± 0.004 C
F_12_	7.21 ± 0.03 C	32,537 ± 140 NC	16,213 ± 204 NC	22 ± 3 NC	33.7 ± 1.2 NC	17,935 ± 592 NC	6977 ± 311 NC	0.39 ± 0.02 NC
F_13_	8.0 ± 1.2 NC	22,513 ± 873 NC	5799 ± 172 NC	44 ± 11 C	60 ± 10 C	6358 ± 298 NC	1791 ± 75 C	0.282 ± 0.006 C
F_14_	10.9 ± 0.2 NC	60,910 ± 1467 NC	24,680 ± 549 NC	30.9 ± 0.2 NC	45.8 ± 0.4 NC	27,372 ± 1399 NC	9645 ± 932 NC	0.35 ± 0.03 NC
F_15_	0.98 ± 0.06 NC	440 ± 80 NC	19 ± 3 NC	3.6 ± 0.6 NC	3.6 ± 1 NC	112 ± 8 NC	57 ± 3 NC	0.513 ± 0.016 NC

Key: Control rate (CR); compliant (C); noncompliant (NC).

**Table 7 pharmaceutics-12-00647-t007:** Regression parameters resulting from the different rheological model fitting to the acquired rheological data.

ID	Ostwald de Waele		Herschel-Bulkley		Bingham	
	*τ = K.$\overset{˙}{ɣ}$^n^*	R^2^	*τ = τ* _0_ *+ K.$\overset{˙}{ɣ}$^n^*	R^2^	*τ = τ* _0_ *+ K.$\overset{˙}{ɣ}$*	R^2^
F_1_	4.062.$\dot{ɣ}$^0.4342^	0.9926	1.05 + 3.1.$\dot{ɣ}$^0.4965^	0.9939	4.125 + 0.4404.$\dot{ɣ}$	0.95
F_2_	86.54.$\dot{ɣ}$^0.3104^	0.8258	83.89 + 8.255.$\dot{ɣ}$^0.8938^	0.9291	87.94 + 5.556.$\dot{ɣ}$	0.9275
F_3_	19.46.$\dot{ɣ}$^0.233^	0.9775	3.894 + 15.47.$\dot{ɣ}$^0.2741^	0.9781	19.32 + 0.7574.$\dot{ɣ}$	0.8649
F_4_	48.44.$\dot{ɣ}$^0.3501^	0.9514	38.97 + 11.78.$\dot{ɣ}$^0.6901^	0.9859	49.6 + 3.6.$\dot{ɣ}$	0.9705
F_5_	91.62.$\dot{ɣ}$^0.2211^	0.7604	88.32 + 2.924.$\dot{ɣ}$^1.09^	0.9412	86.35 + 4.071.$\dot{ɣ}$	0.9402
F_6_	9.929.$\dot{ɣ}$^0.2993^	0.991	4.161 + 5.721.$\dot{ɣ}$^0.4169^	0.9955	9.85 + 0.5764.$\dot{ɣ}$	0.9363
F_7_	65.02.$\dot{ɣ}$^0.2385^	0.9223	53.73 + 10.28.$\dot{ɣ}$^0.6643^	0.9792	63.27 +2.877.$\dot{ɣ}$	0.9598
F_8_	46.05.$\dot{ɣ}$^0.3749^	0.9694	32.06 + 16.16$\dot{ɣ}$^0.6235^	0.9886	47.46 + 3.853.$\dot{ɣ}$	0.965
F_9_	39.64.$\dot{ɣ}$^0.4130^	0.9775	23.91 + 18.01.$\dot{ɣ}$^0.6001^	0.9885	41.30 + 9.911.$\dot{ɣ}$	0.9615
F_10_	135.1.$\dot{ɣ}$^0.115^	0.4347	124.8 + 0.9316.$\dot{ɣ}$^1.387^	0.7642	118.8 + 3.861.$\dot{ɣ}$	0.7508
F_11_	57.24.$\dot{ɣ}$^0.2924^	0.9674	35.81 + 21.32.$\dot{ɣ}$^0.5125^	0.9838	56.95 + 3.233.$\dot{ɣ}$	0.941
F_12_	59.34.$\dot{ɣ}$^0.3036^	0.9263	52.65 + 8.325.$\dot{ɣ}$^0.7822^	0.9863	59.09 + 3.679.$\dot{ɣ}$	0.9799
F_13_	60.1.$\dot{ɣ}$^0.3452^	0.9497	49.49 + 13.43.$\dot{ɣ}$^07069^	0.988	61.34 + 4.444.$\dot{ɣ}$	0.975
F_14_	110.1.$\dot{ɣ}$^0.2337^	0.6908	106.3 + 4.432.$\dot{ɣ}$^1.042^	0.8661	105.1 + 5.173.$\dot{ɣ}$	0.8659
F_15_	5.232.$\dot{ɣ}$^0.409^	0.9896	1.316 + 3.997.$\dot{ɣ}$^0.4702^	0.9909	5.298 + 0.5069.$\dot{ɣ}$	0.9402
ID	Casson		Cross			
	*τ^1/2^= τ* _0_ *^1/2^ + (K.$\overset{˙}{ɣ}$)^1/2^*	R^2^	*η = η_∞_ + [(η* _0_ *− η_∞_)/(1+ (C_._$\overset{˙}{ɣ}$)^m^)]*			R^2^
F_1_	2.26^1/2^ + (0.258.$\dot{ɣ}$)^1/2^	0.9838	1.804 + [(2.73e^+04^ *−* 1.804)/(1 + 716.$\dot{ɣ}$)^0.436^)]			0.9945
F_2_	66.39*^1/2^ +* (1.939.$\overset{˙}{ɣ}$)*^1/2^*	0.8762	16.02 + [(25.59e^+04^ *−* 16.02)/(1 + 341.$\dot{ɣ}$)^0.785^)]			0.9978
F_3_	14.3*^1/2^ +* (0.2839.$\overset{˙}{ɣ}$)*^1/2^*	0.9485	3.647 + [(6.99e^+04^ *−* 3.647)/(1 + 1197.$\dot{ɣ}$)^0.212^)]			0.9987
F_4_	35.11^1/2^ + (1.517.$\dot{ɣ}$)^1/2^	0.976	12.27 + [(0.31e^+04^ *−* 12.27)/(1 + 26.54.$\dot{ɣ}$)^0.506^)]			0.998
F_5_	72.23^1/2^ + (1.054.$\dot{ɣ}$)^1/2^	0.8754	11.34 + [(13.35e^+04^ *−* 11.34)/(1 + 335.12.$\dot{ɣ}$)^0.797^)]			0.9926
F_6_	6.818^1/2^ + (0.2492.$\dot{ɣ}$)^1/2^	0.9813	2.073 + [(3.23e^+04^ *−* 2.073)/(1 + 1596.$\dot{ɣ}$)^0.147^)]			0.9974
F_7_	50.06^1/2^ + (0.903.$\dot{ɣ}$)^1/2^	0.9721	9.467 + [(7.087e^+04^ *−* 9.467)/(1 + 557.$\dot{ɣ}$)^0.1568^)]			0.9903
F_8_	31.76^1/2^ + (1.766.$\dot{ɣ}$)^1/2^	0.9818	11.35 + [11.2e^+04^ *−* 11.35)/(1 + 549.$\dot{ɣ}$)^0.342^)]			0.999
F_9_	25.78^/2^ + (1.98.$\dot{ɣ}$)^1/2^	0.9833	12.91 + [(23.51e^+04^ *−* 12.91)/(1 + 610.$\dot{ɣ}$)^0.54^)]			0.997
F_10_	104.2^1/2^ + (0.7152.$\dot{ɣ}$)^1/2^	0.6719	21.34 + [(9.021e^+04^ *−* 21.34)/(1 + 117.43.$\dot{ɣ}$)^0.544^)]			1
F_11_	40.85^1/2^ + (1.3.$\dot{ɣ}$)^1/2^	0.9747	11.44 + [(1.35e^+04^ *−* 11.44)/(1 + 83.$\dot{ɣ}$)^0.497^)]			0.9981
F_12_	44.64^1/2^ + (1.327.$\dot{ɣ}$)^1/2^	0.9687	13.88 + [(0.41e^+04^ *−* 13.88)/(1 + 341.$\dot{ɣ}$)^0.476^)]			0.999
F_13_	43.79^/2^ + (1.818.$\dot{ɣ}$)^1/2^	0.975	15.17 + [(0.516e^+04^ *−* 15.17)/(1 + 33.$\dot{ɣ}$)^0.494^)]			0.9992
F_14_	84.22^1/2^ + (1.472.$\dot{ɣ}$)^1/2^	0.7959	6.826 + [(5.35e^+04^ *−* 6.826)/(1+ 82.$\dot{ɣ}$)^0.561^)]			1
F_15_	3^1/2^ + (0.2874.$\dot{ɣ}$)^1/2^	0.9774	1.83 + [(2.7e^+04^ *−* 1.83)/(1 + 471.3.$\dot{ɣ}$)^0.683^)]			0.9909

Key: Shear stress (*τ*, Pa); consistency index (*K*, Pa.s^n^); flow behavior index (*n*); yield point (*τ*_0_, Pa); shear rate (*$\overset{˙}{ɣ}$*, s^−1^); zero shear viscosity (*η*_0_, Pa.s); infinite shear viscosity (*η_∞_*, Pa.s); Cross time constant (*C*, s); Cross rate constant (*m*).

**Table 8 pharmaceutics-12-00647-t008:** Effect of independent variables on cream release profile. Regression coefficients resulting from the application of Higuchi and Korsmeyer-Peppas mathematical models to the experimental release data. Data are expressed as mean ± SD (n = 3).

ID	Higuchi-c_1_.√t		Korsmeyer-Peppas-*k*.t^c2^			R_6h_ (%)	R_24h_ (%)
	c_1_ (μg/cm^2^/√t)	R^2^	*k* (t^−1^)	c_2_	R^2^		
F_1_	149 ± 3 C	0.98786	2.83 ± 0.15	0.54 ± 0.02 C	0.99225	7.8 ± 0.2 C	15.4 ± 0.5 C
F_2_	158 ± 3 C	0.99034	3.7 ± 0.2	0.49 ± 0.02 C	0.99081	9.1 ± 0.5 C	16.4 ± 0.6 C
F_3_	147 ± 3 C	0.98910	3.05 ± 0.17	0.49 ± 0.02 C	0.98917	7.6 ± 0.3 C	13.7 ± 0.5 C
F_4_	153 ± 4 C	0.97747	4.1 ± 0.2	0.45 ± 0.02 NC	0.98578	9.6 ± 0.9 C	16.1 ± 1.4 C
F_5_	149 ± 2 C	0.99276	3.8 ± 0.2	0.48 ± 0.02 C	0.99224	9.3 ± 0.9 C	17.0 ± 1.2 C
F_6_	112 ± 2 NC	0.98950	2.84 ± 0.17	0.52 ± 0.02 C	0.98932	7.5 ± 0.4 C	14.1 ± 0.9 C
F_7_	125.4 ± 1.6 NC	0.99546	2.06 ± 0.08	0.502 ± 0.015 C	0.99544	5.2 ± 0.0 NC	9.9 ± 0.1 NC
F_8_	114 ± 2 NC	0.99245	2.04 ± 0.11	0.50 ± 0.02 C	0.99125	5.1 ± 0.4 F	9.7 ± 0.8 F
F_9_	137 ± 2 C	0.99316	2.47 ± 0.09	0.481 ± 0.015 C	0.99495	6.0 ± 0.2 F	11.1 ± 0.4 NC
F_10_	173 ± 2 C	0.99489	3.00 ± 0.11	0.523 ± 0.014 C	0.99587	7.9 ± 0.2 C	15.5 ± 0.7 C
F_11_	138 ± 2 C	0.99231	2.39 ± 0.11	0.53 ± 0.02 C	0.99407	6.4 ± 0.6 F	12.5 ± 0.8 NC
F_12_	134 ± 5 C	0.96969	2.67 ± 0.13	0.63 ± 0.02 C	0.99549	8.8 ± 1.7 C	19.3 ± 2.5 C
F_13_	156 ± 2 C	0.99532	2.63 ± 0.09	0.506 ± 0.015 C	0.99551	6.8 ± 0.3 C	12.9 ± 0.9 NC
F_14_	157 ± 3 C	0.99226	3.08 ± 0,16	0.50 ± 0.02 C	0.99076	8.3 ± 1.2 C	14.7 ± 1.2 C
F_15_	196 ± 7 C	0.96899	4.1 ± 0.4	0.59 ± 0.03 C	0.98454	12.4 ± 0.2C	25 ± 1 C

Key: Compliant (C); noncompliant (NC).

**Table 9 pharmaceutics-12-00647-t009:** Effect of independent variables on cream permeation profile. Permeation parameters according to experimental permeation data. Data are expressed as mean ± SD (n = 3).

ID	J_ss_ (μg/cm^2^/h)	ER (J_ss_)	K_p_ (e^−2^) (cm/h)	Q_6h_ (μg/cm^2^)	Q_24h_ (μg/cm^2^)	Q_48h_ (μg/cm^2^)	t_lag_ (h)
F_1_	0.27 ± 0.04 NC	0.77	0.839 NC	1.6 ± 0.9 NC	4 ± 2 C	12 ± 7 NC	10
F_2_	0.31 ± 0.09 C	0.89	0.953 NC	1.8 ± 1.1 NC	5 ± 3 C	17 ± 10 C	10
F_3_	0.4 ± 0.2 C	1.14	1.12 C	1.8 ± 1.0 C	4 ± 2 C	13 ± 7 C	24
F_4_	0.37 ± 0.02 C	1.06	0.991 NC	1.9 ± 1.1 C	5 ± 3 C	16 ± 9 C	10
F_5_	0.43 ± 0.03 C	0.94	1.08 C	1.7 ± 1.0 NC	5 ± 3 C	18 ± 11 C	10
F_6_	0.31 ± 0.11 C	0.89	0.747 NC	1.8 ± 1.0 C	3.0 ± 1.7 C	11 ± 6 NC	24
F_7_	0.49 ± 0.09 C	1.40	1.25 C	1.8 ± 1.0 C	6 ± 3 C	21 ± 12 C	10
F_8_	0.5 ± 0.2 C	1.51	1.41 C	1.7 ± 1.0 NC	4 ± 2 C	17 ± 10 C	24
F_9_	0.43 ± 0.15 C	1.23	1.09 C	1.8 ± 1.1 F	5 ± 3 C	18 ± 11 C	10
F_10_	0.39 ± 0.11 C	1.10	1.01 NC	1.8 ± 1.0 C	5 ± 3 C	17 ± 10 C	10
F_11_	0.97 ± 0.08 C	2.77	2.55 C	1.8 ± 1.1 NC	9 ± 5 C	40 ± 23 C	10
F_12_	0.16 ± 0.03 NC	0.46	0.542 NC	1.7 ± 1.0 NC	3 ± 2 NC	8 ± 5 NC	10
F_13_	0.255 ± 0.015 NC	0.73	0.563 NC	1.6 ± 0.9 NC	3 ± 2 NC	11 ± 7 NC	10
F_14_	0.394 ± 0.142 C	1.13	1.01 NC	1.9 ± 1.1 C	5 ± 3 C	17 ± 10 C	10
F_15_	0.245 ± 0.061 NC	0.70	0.855 NC	1.8 ± 1.0 C	4 ± 2 C	11 ± 6 NC	10

Key: Compliant (C); noncompliant (NC).

**Table 10 pharmaceutics-12-00647-t010:** Microstructure effect on the meaningful cream performance CQAs.

CQAs	IVRT				IVPT				
	c_1_	c_2_	R_6h_	R_24h_	J_ss_	K_p_	Q_6h_	Q_24h_	Q_48h_
Droplet size	+++	+	+++	+++	+	+	+	+	+
*η* _10_	+++	+	+++	+++	+	+	+	+	+
S_R_	+++	+	+++	+++	+	+	+	+	+
LVR plateau	++	+	+	+	+	+	+	+	+
*τ* _0_	++	+	+	+	+	+	+	+	+
*τ_f_*	++	+	+	+	+	+	+	+	+
*G’*	++	+	++	++	+	+	+	+	+
*G″*	++	+	++	++	+	+	+	+	+
*tan δ*	++	+	+	+	+	+	+	+	+
Assay	+++	+++	+++	+++	+	+	+	+	+
pH	++	++	++	++	+++	+++	+++	+++	+++
Instability index	++	+	++	++	+	++	++	++	++
Sedimentation rate	++	+	++	++	+	++	++	++	++
Creaming rate	++	+	++	++	+	++	++	++	++

Key: +, low effect; ++, medium effect; +++, high effect.

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
