# Peer review of "Progressing Towards the Sustainable Development of Cream Formulations"

_pharmaceutics, 2020, doi:10.3390/pharmaceutics12070647_

Round 1
Reviewer 1 Report
My report:
Manuscript ID: pharmaceutics-857733
Title: Progressing Towards the Sustainable Development of Cream Formulations
Authors: Ana Simões, Francisco Veiga and Carla Vitorino
Overview and general recommendation:
The study provides assumptions to assist the sustainable development of cream formulation. The study aimed to rationalize and predict the effect of formulation and process variables on a 1% hydrocortisone cream quality profile, interplaying microstructural properties with product performance and stability. Quality by Design approach was used for data analysis considering a three-factor, three-level Box-Behnken design.
Overall, I found the aim of the study is good and the introduction provides sufficient background and includes relevant references. The study reported the methods and the results are adequately and accurately described and showed very interesting and well organized data. However, there are minor issues that need to be addressed before this manuscript can be deemed publishable.
Comments to the authors:
- The abstract should compromise some of the promising study results.
- In the introduction line 88, the abbreviation between brackets (FMECA) is not defined and It does not represent the name mentioned before it. I think it stands for “Failure mode effects and criticality analysis”. Please revise and confirm.
- Line 132, delete the letter S in the word “DoEs”. Because the plural belongs to the word after it.
- Line 203, what is the volume of distilled water that used for dispersion of the sample before the determination of pH?.
- Line 530, correct the typo mistake in the word “concertation”.
Author Response
Dear Editor,
The authors appreciated the careful review and greatly acknowledge the comments that the Reviewers have provided on the manuscript entitled “Progressing Towards the Sustainable Development of Cream Formulations” that we have submitted for publication in Pharmaceutics. As you will see in the attached files of the manuscript, we have made the recommended changes and answered in detail to the questions raised. All changes made to the text are marked as track-changes. We would greatly appreciate if you accept this revised version of the manuscript.
1st Reviewer:
My report:
Manuscript ID: pharmaceutics-857733
Title: Progressing Towards the Sustainable Development of Cream Formulations
Authors: Ana Simões, Francisco Veiga and Carla Vitorino
Overview and general recommendation:
The study provides assumptions to assist the sustainable development of cream formulation. The study aimed to rationalize and predict the effect of formulation and process variables on a 1% hydrocortisone cream quality profile, interplaying microstructural properties with product performance and stability. Quality by Design approach was used for data analysis considering a three-factor, three-level Box-Behnken design.
Overall, I found the aim of the study is good and the introduction provides sufficient background and includes relevant references. The study reported the methods and the results are adequately and accurately described and showed very interesting and well organized data. However, there are minor issues that need to be addressed before this manuscript can be deemed publishable.
Comments to the Authors:
- The abstract should compromise some of the promising study results.
Response: Results of the most impacted responses by microstructural properties are now provided. Please see “Abstract” section.
- In the introduction line 88, the abbreviation between brackets (FMECA) is not defined and It does not represent the name mentioned before it. I think it stands for “Failure mode effects and criticality analysis”. Please revise and confirm.
Response: FMECA abbreviation was deleted from the “Introduction” section and it is defined in “Methods” section. Please consult line 123.
- Line 132, delete the letter S in the word “DoEs”. Because the plural belongs to the word after it.
Response: The word “DoEs” was corrected accordingly. Please see line 140.
- Line 203, what is the volume of distilled water that used for dispersion of the sample before the determination of pH?.
Response: For pH determination, cream samples was dispersed in 10 mL of distilled water. This procedure was made clearer, please see line 192 and 193.
- Line 530, correct the typo mistake in the word “concertation”.
Response: The word “concertation” was corrected accordingly to “concentration”. Please see line 544.
Reviewer 2 Report
In my opinion, the manuscript entitled "Progressing Towards the Sustainable Development of Cream Formulations" is well written and suitable for this Journal.
This manuscript has merit since presents interesting topic about the development of cream formulations and is useful for the pharmaceutical industry. So, the subject is suitable for publication in this journal after corrections as following:
- Too many abbreviations are used throughout the manuscript. Sometimes it becomes even confusing.
- Line 216: Replace "Rheological Aspects" by "Rheological characterization" or "Rheological measurements"
- Lines 223-224: Specify the geometry used.
- Line 228: Not all oscillating tests are non-destructive, such as stress sweeps.
- Figure 2: These flow curves must be represented in log-log scale in order to aprecciate diferences at low shear rates.
- Concernig this sentence: "The K gives an indication about sample viscosity. However, to be properly compared, the materials should present similar n [5,6,54-58]." You can compare between samples if you modify the model. For example, see references:
- Trujillo-Cayado, L. A., Natera, A., García González, M. D. C., Muñoz, J., & Alfaro Rodríguez, M. D. C. (2015). Rheological properties and physical stability of ecological emulsions stabilized by a surfactant derived from cocoa oil and high pressure homogenization.
- Trujillo-Cayado, L. A., Santos, J., Alfaro, M. C., Calero, N., & Muñoz, J. (2016). A further step in the development of oil-in-water emulsions formulated with a mixture of green solvents. Industrial & Engineering Chemistry Research, 55(27), 7259-7266.
The value of K matches the viscosity (or stress) value at 1s-1.
Author Response
Dear Editor,
The authors appreciated the careful review and greatly acknowledge the comments that the Reviewers have provided on the manuscript entitled “Progressing Towards the Sustainable Development of Cream Formulations” that we have submitted for publication in Pharmaceutics. As you will see in the attached files of the manuscript, we have made the recommended changes and answered in detail to the questions raised. All changes made to the text are marked as track-changes. We would greatly appreciate if you accept this revised version of the manuscript.
2nd Reviewer:
In my opinion, the manuscript entitled "Progressing Towards the Sustainable Development of Cream Formulations" is well written and suitable for this Journal.
This manuscript has merit since presents interesting topic about the development of cream formulations and is useful for the pharmaceutical industry. So, the subject is suitable for publication in this journal after corrections as following:
- Too many abbreviations are used throughout the manuscript. Sometimes it becomes even confusing.
Response: A list of abbreviations is now included in the manuscript.
- Line 216: Replace "Rheological Aspects" by "Rheological characterization" or "Rheological measurements"
Response: "Rheological Aspects" was replaced by "Rheological characterization". Please see lines 204 and 489.
- Lines 223-224: Specify the geometry used.
Response: The oscillatory measurements (amplitude sweep tests and the frequency sweep tests) were carried resorting to the same geometry configuration (plate-plate), with the same geometries: an upper plate (P20/Ti, 20 mm diameter) and a lower plate (TMP 20). This information was added to the text.
Please see lines 236-241.
- Line 228: Not all oscillating tests are non-destructive, such as stress sweeps.
Response: The assumption of rotational measurements as destructive tests and oscillatory measurements as non-destructive tests was corrected. Please consult lines 215 and 216.
- Figure 2: These flow curves must be represented in log-log scale in order to appreciate differences at low shear rates.
Response: In cream viscosity curves, major differences were observed at higher shear rate values. For that reason, we chose to maintain the logarithmic-linear scale representation. However, this information was added in the manuscript. Please consult lines 226 and 227.
Figure 2 in log-lin scale representation:
Figure 2 in log-log scale representation:
- Concerning this sentence: "The K gives an indication about sample viscosity. However, to be properly compared, the materials should present similar n [5,6,54-58]." You can compare between samples if you modify the model. For example, see references:
- Trujillo-Cayado, L. A., Natera, A., García González, M. D. C., Muñoz, J., & Alfaro Rodríguez, M. D. C. (2015). Rheological properties and physical stability of ecological emulsions stabilized by a surfactant derived from cocoa oil and high pressure homogenization.
- Trujillo-Cayado, L. A., Santos, J., Alfaro, M. C., Calero, N., & Muñoz, J. (2016). A further step in the development of oil-in-water emulsions formulated with a mixture of green solvents. Industrial & Engineering Chemistry Research, 55(27), 7259-7266.
Response: In the suggested references it is possible to find a relation between consistency index (K) and power law index (n) values. A direct comparison of formulations K for samples with different n values should be avoided unless a modified power law equation is used. Additional information and references are now provided in the manuscript. Please see lines 587-589.
“References” section must be updated.
- The value of K matches the viscosity (or stress) value at 1s-1.
Response: “K” is the consistency index (Pa.sn) related to the formulation viscosity as detailed in the following references:
- Kim, J.Y.; Song, J.Y.; Lee, E.J.; Park, S.K. Rheological properties and microstructures of Carbopol gel network system. 2003; Vol. 281, pp 614-623.
- Głowińska, E.; Datta, J. A mathematical model of rheological behavior of novel bio-based isocyanate-terminated polyurethane prepolymers. Industrial Crops & Products 2014, 60, 123-129, doi:10.1016/j.indcrop.2014.06.016.
- Ghica, M.V.; Hirjau, M.; Lupuleasa, D.; Dinu-Pirvu, C.-E. Flow and Thixotropic Parameters for Rheological Characterization of Hydrogels. 2016; Vol. 21.

Reviewer 3 Report
The manuscript aimed to discuss how the pharmaceutical development of a topical cream should be carried out based on the last upgrade of the regulatory framework. Indeed, in agreement with the last EMA draft on quality of topical dosage forms a tripartite analysis was adopted, combined the definition of QTPP and CQAs and DoE. The article is beyond doubt very interesting. However, some minor revisions should be done.
The introduction should be upgraded including reference to the current discussion on the implementation of the regulatory framework on the assessment of the quality of topical dosage forms. Additional references to the EU and US regulatory frameworks may be useful to contextualize the approach proposed in the manuscript.
The method used for evaluating the integrity of the skin membrane should be reported in the manuscript.
Considering that fitted models with a Prob>F > 0.05 showed absence of statistically significance, their elimination from Table S2 is suggested to avoid misleading interpretations by the readers. Moreover, the authors are invited to formulate an explanation of the reduced impact of the DoE factors on such responses.
In the results and discussion section, a lot of results were reported and discussed. Maybe, their discussion can be enhanced by underlining better the relationship between factors and responses and the biorelevance of the results obtained by the tripartite analysis.
Line 385. A reference should be added
Lines 396-397 The sentence should be better explained.
The correlation between factors and microstructure variable on the product performance (IVRT; IVPT) should be better discussed in comparison to the existing literature.
The legends of tables and figures should be reviewed to be more descriptive of the table contents (e.g., Figure 8). Moreover, the abbreviation used should be harmonized throughout the manuscript (e.g., F in Table 5).
The structure of Table S2 should be reviewed. In the current version, some data are out of the page.
Author Response
Dear Editor,
The authors appreciated the careful review and greatly acknowledge the comments that the Reviewers have provided on the manuscript entitled “Progressing Towards the Sustainable Development of Cream Formulations” that we have submitted for publication in Pharmaceutics. As you will see in the attached files of the manuscript, we have made the recommended changes and answered in detail to the questions raised. All changes made to the text are marked as track-changes. We would greatly appreciate if you accept this revised version of the manuscript.
3rd Reviewer:
The manuscript aimed to discuss how the pharmaceutical development of a topical cream should be carried out based on the last upgrade of the regulatory framework. Indeed, in agreement with the last EMA draft on quality of topical dosage forms a tripartite analysis was adopted, combined the definition of QTPP and CQAs and DoE. The article is beyond doubt very interesting. However, some minor revisions should be done.
- The introduction should be upgraded including reference to the current discussion on the implementation of the regulatory framework on the assessment of the quality of topical dosage forms. Additional references to the EU and US regulatory frameworks may be useful to contextualize the approach proposed in the manuscript.
Response: Additional references addressing EU and US regulatory frameworks are now provided in the introduction. Please see lines 88 and 93. “References” section must be updated.
Chang, R.K.; Raw, A.; Lionberger, R.; Yu, L. Generic Development of Topical Dermatologic Products: Formulation Development, Process Development, and Testing of Topical Dermatologic Products. AAPS JOURNAL 2013, 15, 41-52.
Osborne, D.W. Impact of Quality by Design on Topical Product Excipient Suppliers, Part I: A Drug Manufacturer's Perspective. Pharmaceutical Technology Europe 2016, 28, 30-37.
Fowler, M. Quality by Design (QbD) Approach to Generic Transdermal or Topical Product Development. American Pharmaceutical Review 2015.
FDA. Guidance for Industry: Transdermal and Topical Delivery Systems - Product Development and Quality Considerations. FDA: USA, 2019
- The method used for evaluating the integrity of the skin membrane should be reported in the manuscript.
Response: Skin membrane integrity analysis was performed by measuring the transepidermal water loss (TEWL). This information is now provided in the manuscript. An additional reference was also considered. Please see lines 281-283.
Zsikó, S.; Csányi, E.; Kovács, A.; Budai-Szűcs, M.; Gácsi, A.; Berkó, S. Methods to Evaluate Skin Penetration In Vitro. Scientia Pharmaceutica 2019, 87, 1-21, doi:10.3390/scipharm87030019.
- Considering that fitted models with a Prob>F > 0.05 showed absence of statistically significance, their elimination from Table S2 is suggested to avoid misleading interpretations by the readers. Moreover, the authors are invited to formulate an explanation of the reduced impact of the DoE factors on such responses.
Response: Considering cream formulations development and optimization, despite the non-significance of some mentioned DoE factors, a detailed understanding of how those factors influence the considered responses, in a synergistic or antagonistic way, will be fundamental to plan future DoE designs. In a previous study, two exhaustive screening designs were conducted in order to select the most important CPPs and CMAs for further optimization. Explanations concerning the non-significance of the screened factors were provided. Please see the reference Simões, A.; Veiga, F.; Vitorino, C. Developing Cream Formulations: Renewed Interest in an Old Problem. Journal of Pharmaceutical Sciences 2019, doi:10.1016/j.xphs.2019.06.006.
- In the results and discussion section, a lot of results were reported and discussed. Maybe, their discussion can be enhanced by underlining better the relationship between factors and responses and the biorelevance of the results obtained by the tripartite analysis.
Response: A summary of the tripartite analysis is available in Table 10 and the main relationships between factors and responses are now described in the section 4.2.7. Summing-up. Please consult lines from 947 to 965.
- Line 385. A reference should be added
Response: The statement refers to a previous screening work performed by the authors. Reference was added. Please see line 390.
Simões, A.; Veiga, F.; Vitorino, C. Developing Cream Formulations: Renewed Interest in an Old Problem. Journal of Pharmaceutical Sciences 2019, doi:10.1016/j.xphs.2019.06.006.
- Lines 396-397 The sentence should be better explained.
Response: An overview of the fitted models indicates that glycerol monostearate amount (x1), homogenization rate (x3) and isopropyl myristate amount (x2) demonstrated a decreasing influence on formulation microstructure, performance and stability. The major and minor impacted CQAs are mentioned in the following two paragraphs. Please consult line 401-408.
- The correlation between factors and microstructure variable on the product performance (IVRT; IVPT) should be better discussed in comparison to the existing literature.
Response: Please see response to Question #4. Also, additional considerations were added, please see lines 771-775.
- The legends of tables and figures should be reviewed to be more descriptive of the table contents (e.g., Figure 8). Moreover, the abbreviation used should be harmonized throughout the manuscript (e.g., F in Table 5).
Response: Figure 8 caption was revised and captions of Tables 7, 8, 9 and 10 were updated. Furthermore, the corresponding “F” abbreviation was harmonized throughout the different tables. Please consult Table 5, 8 and 9.
Figure 8. Design space of glycerol monostearate content, isopropyl myristate content and homogenization rate, comprising the overlay of (A), (B) and (C) optimal regions.
Table 7. Regression parameters resulting from the different rheological model fitting.
Table 8. Effect of independent variables on cream release profile. Regression coefficients resulting from the application of Higuchi and Korsmeyer-Peppas mathematical models to the experimental release data. Data are expressed as mean ± SD (n=3).
Table 9. Effect of independent variables on cream permeation profile. Permeation parameters according to experimental permeation data. Data are expressed as mean ± SD (n=3).
Table 10. Microstructure effect on the meaningful cream performance CQAs.
- The structure of Table S2 should be reviewed. In the current version, some data are out of the page.
Response: Table S2 was restructured.